# Axion topology in photonic crystal domain walls

Chiara Devescovi [1,2,8] ✉, Antonio Morales-Pérez [2,3,8] ✉, Yoonseok Hwang [4], Mikel García-Díez[2,5], Iñigo Robredo[2,6], Juan Luis Mañes[5], Barry Bradlyn [4], Aitzol García-Etxarri [2,7] ✉ & Maia G. Vergniory[2,6] ✉

Axion insulators are 3D magnetic topological insulators supporting hinge states and quantized magnetoelectric effects, recently proposed for detecting dark-matter axionic particles via their axionic excitations. Beyond theoretical interest, obtaining a photonic counterpart of axion insulators offers potential for advancing magnetically-tunable photonic devices and axion haloscopes based on axion-photon conversion. This work proposes an axionic 3D phase within a photonic setup. By building inversion-symmetric domain-walls in gyrotropic photonic crystals, we bind chiral modes on inversion-related hinges, ultimately leading to the realization of an axionic channel of light. These states propagate embedded in a 3D structure, thus protected from radiation in the continuum. Employing a small external gyromagnetic bias, we transition across different axionic mode configurations, enabling effective topological switching of chiral photonic fibers. While demonstrating the possibility of realizing axion photonic crystals within state-of-the-art gyrotropic setups, we propose a general scheme for rendering axion topology at domain walls of Weyl semimetals.

Axion insulators (AXIs)[1–19] are 3D inversion ($\mathcal{I}$)-symmetric magnetic Higher-Order Topological Insulators (HOTIs)[20,21] which induce various topological magnetoelectric effects, such as the quantized magneto-optical Faraday and Kerr rotation, the image magnetic monopole effect, and half-quantized surface Hall conductance[7–10]. The topological properties of AXIs arise from the quantization of their electro-magnetic coupling term, the so-called topological $\theta$-angle[1–3], which is pinned to $\pi$ in presence of $\mathcal{I}$-symmetry (or other $\theta$-odd operations such as roto-inversions and time-reversed rotations)[22].

AXIs are of significant interest because, acting as HOTI, they are able to support hinge-localized chiral modes, which propagate in the form of unidirectional axionic channels[23–25]. These hinge-states are expected to emerge at the 1D facets of an AXI crystallite or in the presence of 1D dislocations in the AXI lattice, where gradients of the $\theta$ angle arise, inducing the formation of axionic strings[10,24,26]. Recent studies[10,16] have shown that the chiral propagation of the AXI hinge-localized modes is highly tunable. Especially in the presence of a fer-romagnetic order, it is possible to switch between different hinge-mode configurations via external magnetic control, allowing magnetic re-routing of conducting channels from one input into one or more outputs. In the context of Photonic Crystals (PhCs), this remarkable property of AXIs could allow to manipulate, direct and deviate the 1D non-reciprocal flow of light, with relevant applications for optical communication technologies and for the development of magnetically-tunable photonic switch devices. So far, no proposals have been presented for axion-based PhCs or axion-protected light

¹Institute for Theoretical Physics, ETH Zurich, 8093 Zürich, Switzerland. ²Donostia International Physics Center, Paseo Manuel de Lardizabal 4, 20018 Donostia-San Sebastian, Spain. ³Material and Applied Physics Department, University of the Basque Country (UPV/EHU), Donostia-San Sebastian, Spain. ⁴Department of Physics, University of Illinois at Urbana-Champaign, Urbana, IL, USA. ⁵Physics Department, University of the Basque Country (UPV/EHU), Bilbao, Spain. ⁶Max Planck Institute for Chemical Physics of Solids, D-01187 Dresden, Germany. ⁷IKERBASQUE, Basque Foundation for Science, María Díaz de Haro 3, 48013 Bilbao, Spain. ⁸These authors contributed equally: Chiara Devescovi, Antonio Morales-Pérez. ✉e-mail: cdevescovi@phys.ethz.ch; antonio.morales@dipc.org; aitzolgarcia@dipc.org; maia.vergniory@cpfs.mpg.de

propagation. Furthermore, recent studies have suggested the use of AXI materials for detecting axion-like particles, that constitute dark-matter candidates[27–29]. This is due to the fact that emergent axionic excitations in AXI couple with electromagnetism, $\mathcal{L} \propto \theta \mathbf{E} \cdot \mathbf{B}$, similar to the axion-photon coupling observed in high-energy physics for light dark-matter, which follows $\mathcal{L} \propto a\mathbf{E} \cdot \mathbf{B}$. In PhCs, photons can interact with external magnetic fields via gyrotropy, they display a non-zero effective mass, and they are wavelength-tunable via lattice size-scaling, all of which are essential ingredients for the realization of an axion haloscope[27,30–32]. The demonstration of an AXI in a PhC could represent an opportunity to bridge these two different approaches in the study of axion-photon coupling.

Despite the theoretical significance and potential applications of AXIs, no proposals have been put forward yet for their implementation in PhCs: our work aims to propose and demonstrate the first theoretical model and general design strategy for photonic AXIs in 3D PhCs. To induce a photonic axionic band topology, we incorporate a phase obstruction in the Supercell Modulation (SM) of the dielectric elements within gyrotropic Weyl PhCs[33,34]. The SM is designed as an $\mathcal{I}$-symmetric, static, geometric deformation of the PhC lattice, enabling an experimental implementation of the PhC without necessitating any dynamic driving. Serving as a photonic analog of a Charge-Density-Wave (CDW)[25,35–37], the SM couples Weyl points with opposite topological charges while maintaining the $\mathcal{I}$-symmetry of the model.

The resulting AXI is dubbed relative because it is only exhibited at the interface of two PhCs with a quantized relative axionic angle $\delta\theta$ and vanishing relative Chern numbers. This approach is grounded in the concept that a dislocation of the CDW phase in a specific class of $\mathcal{I}$-symmetric Weyl Semimetals (WS) acts as a dynamic axion field[25,26,35]. Consequently, the domain wall separating the phase-obstructed CDW-WS can be interpreted as the critical point between a trivial insulator and an AXI. By employing this strategy, we successfully realize a photonic relative Axion Insulator (rAXI) in a realistic gyrotropic setup.

By inserting planar dislocations in the SM phase, we bind 1D chiral channels on $\mathcal{I}$-related hinges, that provide a PhC realization of an axion domain wall protected by $\mathcal{I}$-symmetry.

Remarkably, the 1D channels supported by the PhC are buried in a fully connected 3D dielectric structure, thus protected from radiation through the electromagnetic continuum[38]. This design not only represents the first instance of a tunable HOTI with chiral hinge states in 3D PhCs[39], but the observed 1D-modes are also consistent with a single, unidirectional axionic channel that wraps around the central phase-obstructed core, endowing the photonic hinge-channels with non-reciprocal propagation properties.

Lastly, we propose a physically viable method for manipulating these axionic hinge modes by controlling the PhC gyrotropic response using a small external magnetic bias. Specifically, we induce gyrotropy-induced transitions in the photonic AXI, which function as an efficient topological switch between various 1D photonic fiber configurations. Interestingly, recent experimental advancements in 3D gyrotropic crystals have demonstrated that imparting a magnetic response to 3D photons is possible, with a high degree of control and intensity[40,41]. These findings suggest the possibility of manipulating, directing, and deviating the 1D non-reciprocal flow of light in a photonic AXI using state-of-the-art experimental setups. The capability of manipulating the HOTI hinge states in the photonic AXI via gyrotropy underscores the potential of the proposed design for creating magnetically tunable photonic switch devices, thereby paving the way for advancements in axion-based photonics.

The main body of this manuscript is divided into three sections: In Section "Relative axion topology", we provide the bulk design and topological characterization of the photonic rAXI; In Section "Phase-obstructed domain walls", we show how to make rAXI topology manifest by creating a domain wall between phase-obstructed $\mathcal{I}$-symmetric rAXIs; In Section "Gyrotropy-induced switching of HOTI states", we demonstrate how to generate and manipulate the higher-order topology of the PhC, by controlling the chiral propagation of the axionic channels of light. Lastly, in the "Methods" section, we demonstrate how to efficiently simulate the electromagnetic response of the rAXI using a Transversality-Enforced Tight-Binding (TETB) model[42]. This model is capable of capturing and regularizing the $\Gamma$-point electromagnetic obstruction that arises in 3D PhCs, due to the transversality constraint of the Maxwell equations[42,43].

## Results and discussion

### Relative axion topology

Our starting setup for inducing photonic AXI band topology consists of an $\mathcal{I}$-symmetric gyrotropic PhC[34,42,44] under an external magnetic field $\mathbf{H} = (0, 0, H_z)$, as shown in Fig. 1a.

In the presence of a gyroelectric medium the external magnetic field induces an off-diagonal imaginary component in the permittivity tensor[45–48], as expressed by the following equation:

$$\varepsilon_{\eta_z} = \begin{pmatrix} \varepsilon_\perp & i\eta_z & 0 \\ -i\eta_z & \varepsilon_\perp & 0 \\ 0 & 0 & \varepsilon \end{pmatrix}, \tag{1}$$

where $\eta_z = \eta_z(H_z)$ is the bias-dependent gyroelectric parameter with $\varepsilon_\perp = \sqrt{\varepsilon^2 + \eta_z^2}$ and $\varepsilon$ the dielectric constant (here $\varepsilon = 16$). As a consequence of Time-Reversal Symmetry (TRS) breaking due to gyrotropy, a photonic Weyl dipole is generated in the Brillouin zone, along the direction of the $H_z$ magnetic field, as shown in Fig. 1a. In the case of this dielectric lattice realization, the Weyl dipole separation increases proportionally to the external $H_z$ and can be magnetically tuned.

In order to emulate the effect of a CDW in condensed-matter systems[23,25,36] to open a topological gap, we introduce a z-directed SM of lattice period $N \in \mathbb{N}$ and $N \geq 2$, commensurate with the Weyl dipole separation $\mathbf{Q} = \mathbf{q}_+ - \mathbf{q}_-$, where $\mathbf{q}_\pm$ are the locations of Weyl points with chirality $\pm 1$ in Brillouin zone. Accordingly, we fix the Weyl points of opposite topological charge at approximately $\mathbf{q}_\pm = (\pi, \pi, \pi \pm \pi/N)$. This results in a folding of the BZ, as shown in Fig. 1b, and couples the Weyl points to open a non-trivial gap, as shown in Fig. 1c.

The SM is introduced as a local deformation $\delta r$ of the radius $r$ of the dielectric rods, according to the relation:

$$\delta r = r_m \cos(2\pi z/Na + \phi), \tag{2}$$

where $a$ is the lattice parameter of the starting photonic crystal, while $r_m$ and $\phi$ control, respectively, the amplitude and the phase of the dielectric modulation. Note that this represents a static geometric deformation of the PhC structure which can be stably implemented during the fabrication process and does not require any dynamical driving. The $\phi$ phase of the SM is the fundamental design parameter that we will set to induce axionic band topology.

To preserve the $\mathcal{I}$-symmetry of the unperturbed PhC of Fig. 1a, which is crucial for axion behavior, we pin the modulation at the $\mathcal{I}$-center and target only two specific values of the SM phase: $\phi = 0$ and $\phi = \pi$. The corresponding modulated dielectric structures are shown in Fig. 1c, d in a 3D rendering, and in Fig. 2 in a side view, for a $N = 3$ modulation period. We observe that both $\phi = 0$ and $\phi = \pi$ phases display the same insulating spectrum. However, we will now demonstrate that their 3D photonic bulk gaps exhibit a different topological obstruction in the $\mathcal{I}$-symmetry indicators associated with the quantization of their relative axion angle $\delta\theta$.

To efficiently model the electromagnetic response of the PhC, we develop an analytical model of the 3D photonic bulk bands via the TETB method introduced by ref. 42. The TETB model is constructed via the introduction of $\mathbf{v}^l$ auxiliary longitudinal modes, able to regularize the $\Gamma$-point obstruction arising from the transversality constraint of the

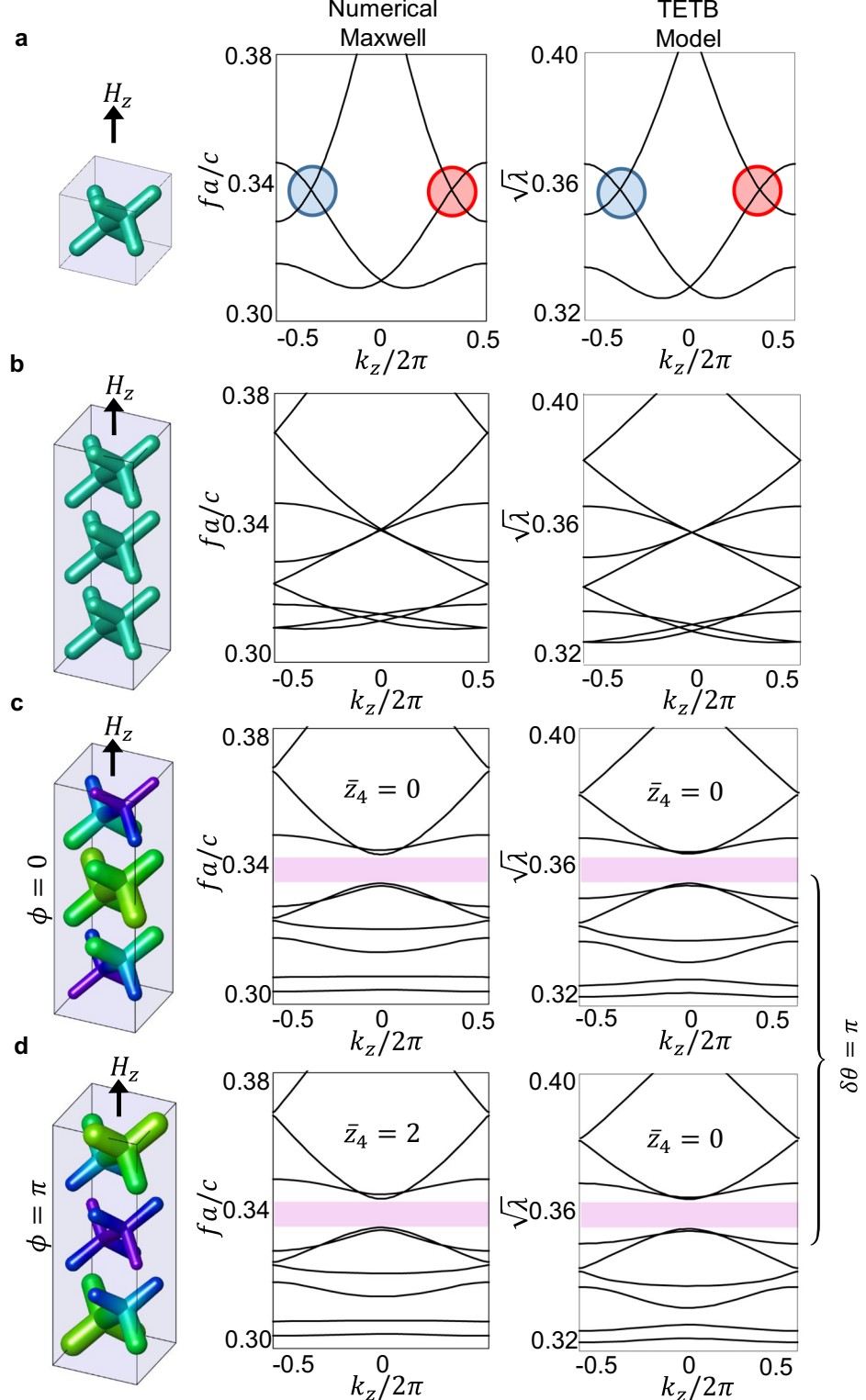

**Fig. 1 | Effect of the angular phase ($\phi$) of the supercell modulation on the photonic bands.** Reduced frequencies ($fa/c$), obtained by solving numerically the Maxwell equations, with $c$ the speed of light and $a$ the lattice parameter, map to the square root of the eigenvalues ($\sqrt{\lambda}$) obtained from the transversality-enforced tight-binding model, consistent with the quadratic mapping $\lambda$ - $\omega^2$. Dispersion shown along $k_z$, for $(k_x, k_y) = (\pi, \pi)$. Weyl points separated by $|\mathbf{Q}| = 2\pi/N$ in momentum space (**a**) overimpose on an artificial supercell (**b**), and then couple by a commensurate modulation of period $N$, leading to gap opening (shaded pink). The supercell amplitude is is $r_m/r_0 = 1/20$ for the crystal and $V_{4c} = -V_{4b} = 1/150$ for the model. The gyrotropic parameter $\eta_z = 7.8$ corresponds to a magnetic perturbation of $H_z = 5.45$. **c**, **d** differ solely for the following angular phase of the supercell modulation: $\phi = 0$ (**c**) and $\phi = \pi$ (**d**).

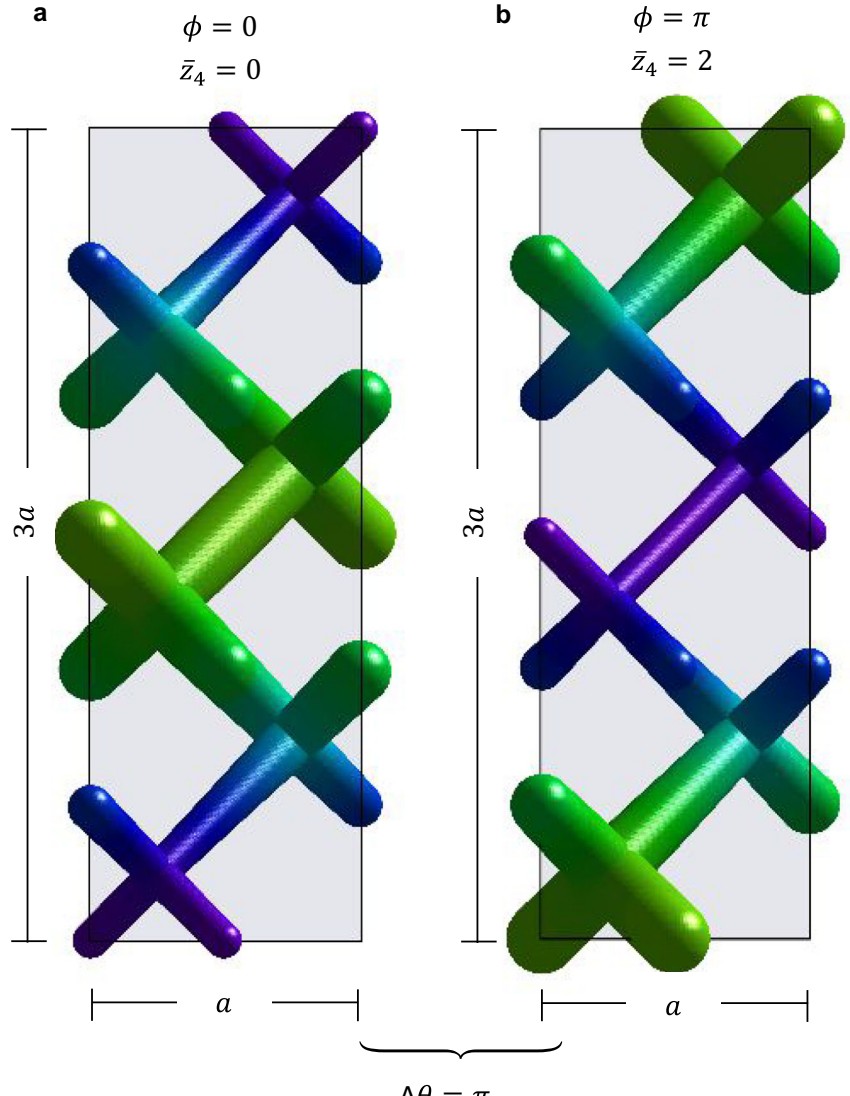

**a** $\phi = 0$ $\bar{z}_4 = 0$

**b** $\phi = \pi$ $\bar{z}_4 = 2$

$3a$

$a$

$3a$

$a$

$\Delta\theta = \pi$

**Fig. 2 | The two phases of the relative axion insulator: side view of the dielectric profile, showing local deformation of the diameter of the dielectric rods, over a period of N = 3.** The z-directed modulation is along the magnetization axis. The modulation is centered at the inversion center of the unperturbed lattice: $\phi = 0$ (**a**) and $\phi = \pi$ (**b**).

Maxwell equations, as proposed in ref. 43. The positive-energy solutions of the TETB are mapped to frequency dispersion of the $\mathbf{v}^T$ transverse electromagnetic modes, obtained by numerically solving the Maxwell equation via the MIT Photonic Bands (MPB) package[49]. The SM is introduced in the TETB via a simple onsite supercell-modulated potential, that mimics the local electromagnetic energy redistribution in the modulated dielectric rods (see "Methods" section):

$$H_\Delta(\mathbf{r}, \mathbf{H}) = H(\mathbf{r}, \mathbf{H}) + \sum_i V_i \cos\left(\frac{2\pi z_i}{Na} + \phi\right) c_i^\dagger(\mathbf{r}) c_i(\mathbf{r}), \quad (3)$$

where $H(\mathbf{r}, \mathbf{H})$ is the real-space TETB Hamiltonian for the magnetic system before modulation, $a$ is the lattice parameter of the crystal before modulation, and $V_i$ and $\phi$ parameterize the amplitude and the phase of the modulation, respectively. Note that the sum in Eq. (3) runs over all the basis pseudo-orbitals used in the TETB model. As shown in the "Methods" section, the TETB reproduces all the bulk properties of

the supercell-modulated PhC; a comparison of the respective bands and topology is displayed in Fig. 1.

To understand the role of $\mathcal{I}$-symmetry in protecting the rAXI topology, we compute the magnetic symmetry-indicators (SI) $\nu_\phi^T = \{\bar{z}_{2,x}, \bar{z}_{2,y}, \bar{z}_{2,z} | \bar{z}_4\}$[18,50–54] for the transverse-electromagnetic modes of the PhC (the $^-$ overbar stands for magnetic and the $()^T$ superscript indicates transverse bands). In particular, we focus our interest on the $\bar{z}_4$ strong index, which is associated to axion topology[17,35,54].

As shown in the "Methods" section, we obtain, correspondingly for the two structures at $\phi = 0$ and $\phi = \pi$:

$$\nu_{\phi=0}^T = \{0, 0, 1 | 0\} \quad (4)$$

and

$$\nu_{\phi=\pi}^T = \{0, 0, 1 | 2\} \quad (5)$$

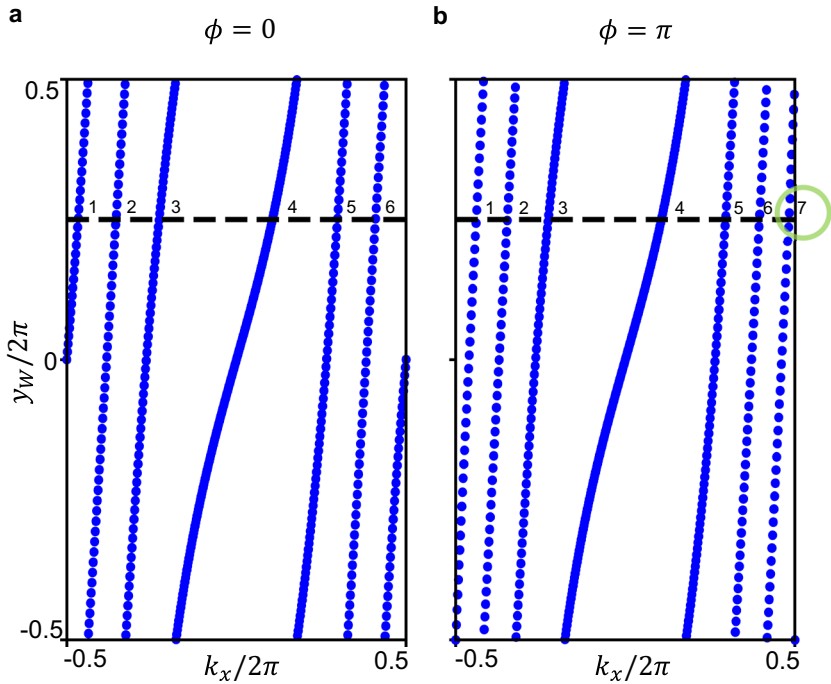

**Fig. 3 | Topological characterization of the AXI surface.** Layer Wilson loop for a $z$-slab at $\phi = 0$ (**a**) and $\phi = \pi$ (**b**) with $n_z = 6$ layers. The $y$ Wannier energy centers wind respectively $n_z$ and $n_z + 1$ times along $k_x$, with the $+1$ discontinuity shown in the green circle. This confirms a $\delta\theta = \delta\phi = \pi$ difference in the relative axion angle.

where the $\delta\bar{z}_4 = 2$ discontinuity of the even $\bar{z}_4$ index stands to indicate a relative axionic obstruction. On the other hand, the invariance of the $\bar{z}_{2,z}$ term is related to an odd $C_z$ Chern invariant, which, as confirmed via photonic Wilson loop[33,55,56] calculations, is $C_z = 1$ identically for both structures. Note that although we have computed $\nu^T_{\phi=0}$ and $\nu^T_{\phi=\pi}$ using the TETB model, the difference:

$$\nu^T_{\phi=\pi} - \nu^T_{\phi=0} = \{0, 0, 0|2\} \qquad (6)$$

depends only on the sign of the modulation-induced band gap. Thus, we find that Eq. (6) holds for the PhC.

To verify the quantization of the relative axion angle between $\phi = 0, \pi$, we compute the layer Chern number $G_z$ of a $z$-slab with its normal along the magnetization axis. As demonstrated in refs. 10,22, a non-zero quantized axionic phase $\theta$ will manifests as an offset in $G_z$, according to the relation:

$$G_z = C_z n_z + \theta/\pi \qquad (7)$$

where $n_z$ counts the layers of the slab. Via this equation, we can extract the $\theta$ axion angle, from the $C_z$ Chern number of a single layer.

The slab Wilson loops, shown in Fig. 3, wind $n_z$ and $n_z + 1$ times, respectively for $\phi = 0$ and $\phi = \pi$, confirming a $\delta\theta = \pi$ discontinuity in the axion $\theta$ angle. Therefore the $\phi = 0, \pi$ supercell-modulated PhC represent rAXI.

### Phase-obstructed domain walls

In this section, our goal is to manifest the relative axion topology. To accomplish this, we create a domain wall in $x$ between the photonic 3D insulator with $\phi = 0$ and its obstructed counterpart with $\phi = \pi$, i.e., imposing a relative axion phase difference of $\delta\theta \equiv \delta\phi = \pi$, as shown in Fig. 4a.

We expect this domain wall configuration to be formally equivalent to the critical point between an AXI with $\theta = \pi$ and a trivial insulator[25,26,26,35,57] and therefore gapped. To ensure a surface gap, we apply a slight tilt to the magnetic field directed towards the $z$-axis, represented as with:

$$\mathbf{H} = (|h| \cos(\sigma), |h| \sin(\sigma), H_z) \qquad (8)$$

and $|h| \ll |H_z|$. As shown in Supplementary Figs. S1–S4, the component of the magnetic perturbation normal to the interface plane ensures the existence of a surface gap, which is essential for the observation of the higher-order topology of the rAXI. The tilted external field couples to the PhC, inducing an in-plane gyrotropic perturbation $\eta_{x,y} = \eta_{x,y}(h_{x,y})$ in the permittivity tensor. As a result, the PhC domain wall bands are gapped, as shown in Fig. 4b.

The size of the surface gap can be controlled via the $h_{x,y}$ bias, by gradually deviating from the gapless condition which results from the boundary condition choice, as demonstrated in the SM. In what follows, we select a boundary condition in which the size of the surface gap disappears in the absence of any magnetization orthogonal to the interface plane: this boundary configuration is reached by maintaining the rod geometry continuously connected across the interface for the PhC. In the TETB, this corresponds to a surface potential that linearly interpolates between the two modulations.

Importantly, the $\phi = 0$ and $\phi = \pi$ structures differ only in their $\bar{z}_4$ index but have an identical Chern vector. It is critical to maintain the condition of equal Chern vectors across the interface in order to prevent anomalous Hall surface states from populating the surface gap, consistently with vectorial bulk-boundary correspondence[34].

### Gyrotropy-induced switching of HOTI states

Next, to generate and manipulate a chiral hinge channel of light, we will be investigating the higher-order topology of the PhC. For this

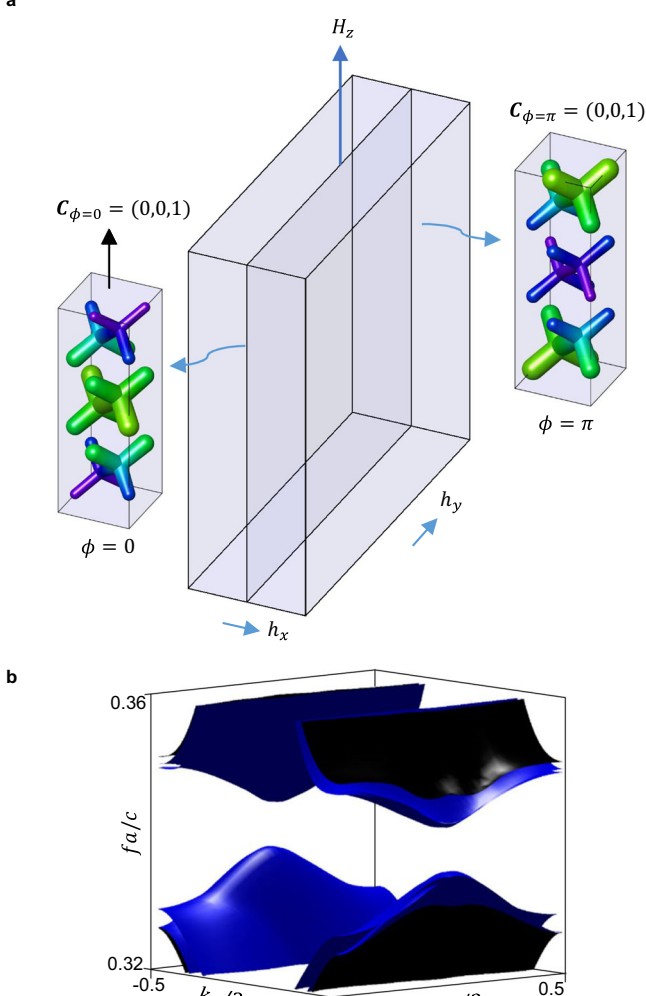

**a**

**Fig. 4 | Axionic surface gap for an $\mathcal{I}$-symmetric domain wall with $\delta\theta \equiv \delta\phi = \pi$.** In (**a**), PhC geometry of the phase-obstructed domain wall configuration. In (**b**), domain wall band structure on the $x = 0$ plane, with projected bulk bands in black, and surface-localized states in blue. Full-wave electromagnetic simulations in MPB.

purpose, we construct an $\mathcal{I}$-symmetric $z$-wire configuration. We embed a $N_x \times N_y$ core of $\phi = 0$ PhC inside a $2N_x \times 2N_y$ region of PhC. Both PhCs are made of the same material, but the latter has $\phi = \pi$. To keep the simulations affordable, we compute the boundary modes for this rod geometry via the use of the TETB model.

As shown in Fig. 5c, chiral gapless modes emerge as in-gap states in the projected domain wall bands, consistent with the bulk-hinge correspondence of the photonic rAXI. The corresponding dielectric structure, which is fully connected, is shown in Fig. 6a, with the central rod extruded upwards, for better visualization. The HOTI states are consistent with the existence of a single unidirectional mode wrapping around a central phase-obstructed core. Moreover, their group velocity can be easily switched by flipping of the external magnetic bias $H_z$. Displayed for a cross-section of the connected structure in Fig. 5a–e, the 1D channels localize on $\mathcal{I}$-related hinges parallel to the $z$ direction.

It is noteworthy that not all of the four $\mathcal{I}$-related hinges support chiral modes at once. Instead, the localization on either a pair of $\mathcal{I}$-related hinges or the other can be chosen by rotation of the small $h_{x,y}$ bias in the $xy$ plane, leading to 4 possible realizations of the hinges, $\alpha, \beta$ (with occupancy of the hinges passing through the corners on the $1\bar{1}0$ diagonal) and $\gamma, \delta$ (with occupancy of the hinges passing through the corners on the 110 diagonal), as shown in Fig. 6b–e. These different

hinge-state configurations are plotted in Fig. 6 at the Γ point for the upwards-moving state. As shown in Supplementary Fig. S3, they can be regarded as distinct boundary-obstructed phases[58,59], since a surface gap (but not a bulk gap) must close in passing from one configuration to another.

The $\alpha,\beta,\gamma,\delta$ gyrotropic-bias-field induced transitions offer a promising and physically accessible way to manipulate the photonic 1D modes, via rotation of the PhC gyrotropic axis through magnetic control by external field. Therefore, the present platform can provide an effective photonic topological switch between different 1D photonic fiber configurations.

Remarkably, the observed hinge modes are embedded within a fully connected 3D dielectric structure, making them highly suitable for guided-light communication applications, as they are protected from radiation through the electromagnetic continuum[38].

By proposing the first tunable HOTI chiral hinge states in PhCs[39], we provide a PhC realization and a distinct manifestation of the axionic hinge states predicted in supercell-modulated Weyl semimetals[24–26,35,57].

More specifically, the hinge modes of Fig. 5b–e are consistent with the presence of a single, unidirectional axionic mode wrapping around a central phase-obstructed core[60].

In conclusion, we proposed a novel design strategy to induce axionic band topology in a gyrotropic PhC and demonstrated the potential use for magnetically tunable photonic switch devices. This approach provides a realistic and physically accessible platform for generating and manipulating the higher-order topology of the AXI PhC, enabling effective topological switching between different configurations for axionic hinges of light. In addition to its fundamental theoretical significance, related to the possibility of coupling between photonic axionic excitations and dark-matter axions, the realization of AXI PhC has the potential to open up the field of axion-based topology, enabling more efficient and versatile control of light propagation in photonic crystals, and thus advancing the state-of-the-art in photonic communication and optical technologies.

## Methods
### Transversality-enforced tight-binding model for a supercell-modulated PhC
To efficiently simulate the electromagnetic response of the photonic rAXI, we develop a TETB model[42], capable of capturing and regularizing the Γ-point electromagnetic obstruction that arises due to the transversality constraint of the Maxwell equations[42,43]. The method allows the demonstration of HOTI topology[61–64], by investigating the bulk-hinge correspondence of the rAXI through cost-effective calculations of large-scale slab- and rod geometries. Please note that differently from other methods tailored for studying boundary responses in finite photonic systems, such as coupled-dipole approaches[65,66], our method exclusively utilizes space-group symmetry principles, paralleling the tight-binding models in solid-state physics. Our approach further incorporates Maxwell's transversality constraints to ensure accurate electromagnetic modeling. The methods detailed in this section, applied herein for the specific PhC of interest, are broadly applicable to any unit cell (UC) and any of the 230 space groups (SGs), as outlined in ref. 42. This generalization remains valid even in the presence of (gyro-)magnetism within the PhC structure[42]. The underlying PhCs that constitute the starting point of our rAXI design are the gyrotropic Weyl semimetals shown in Fig. 1a. Before the introduction of the $z$-directed external magnetic field and SM, the crystal structure belongs to SG #224 ($Pn\bar{3}m$)[33,34,42].

The symmetry content of the photonic bands can be deduced by analyzing the Bloch electric modes (**E**), obtained in MPB via numerical solution of the Maxwell equations. The **E** field transforms as a vector:

$$g\mathbf{E}(\mathbf{r}) = (R\mathbf{E})(R^{-1}(\mathbf{r} - \mathbf{t})), \tag{9}$$

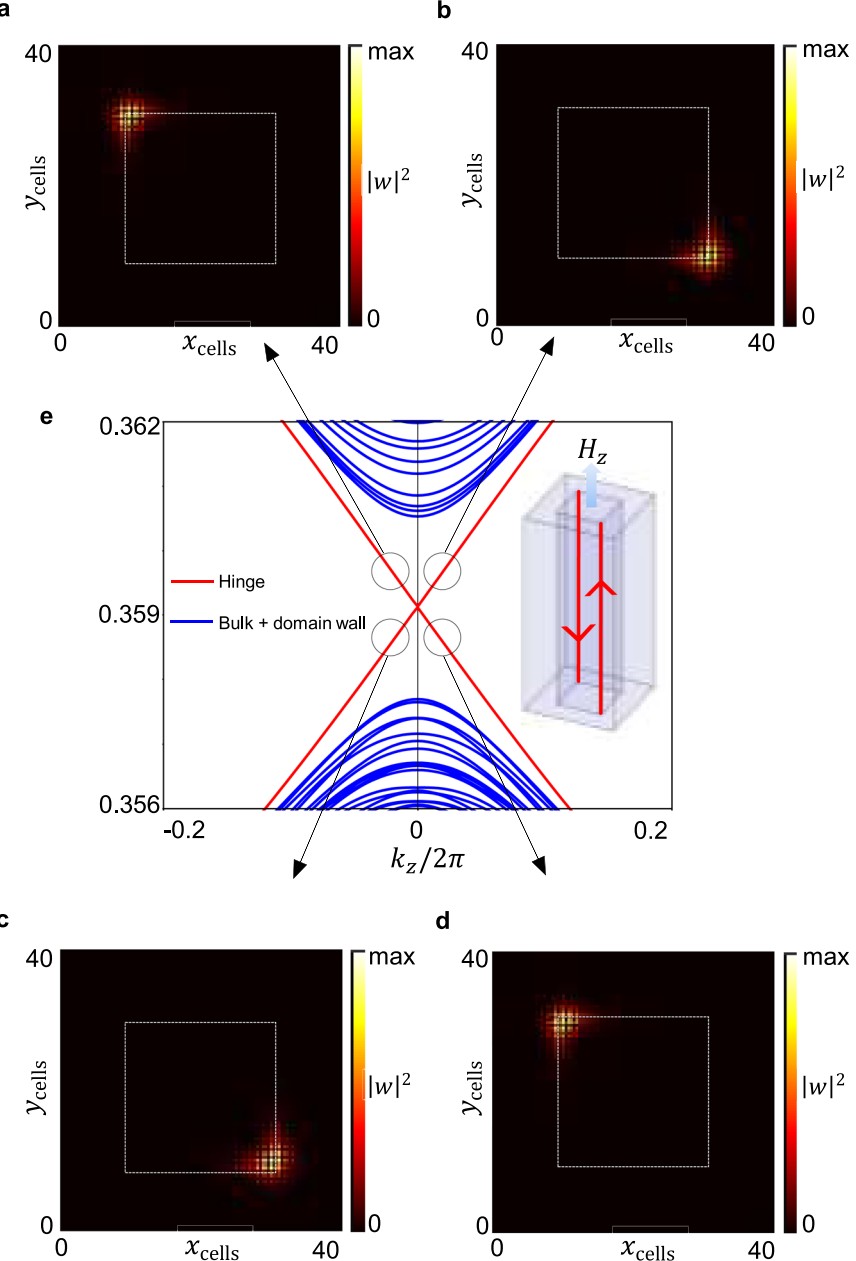

**Fig. 5 | Gapless AXI hinge states evaluated in a *z*-wire configuration, with $2N_x \times 2N_y = 40 \times 40$ cells.** The crystal structure is fully connected but presents an axion phase discontinuity of $\delta\theta \equiv \delta\phi = \pi$. Projected surface bands in blue, hinge bands in red, in (**c**). The chiral modes are localized on $\mathcal{I}$-related hinges: a *xy*-cross-section of the *z*-wire geometry is shown in (**a**, **b**, **d**, **e**). The flipping of the external $H_z$ field results in an overall exchange of the group velocity signs. These HOTI states are consistent with the existence of a single unidirectional mode wrapping around a central phase-obstructed core.

for each space group operation $g = \{R|\mathbf{t}\}$, where $R$ is a point group element and $\mathbf{t}$ a translation.

For each band $n$ with $\omega \neq 0$ and every high-symmetry point $\mathbf{k}_h$, we compute the $x_{n,\mathbf{k}_h}(g)$ diagonal elements of the representation matrix corresponding to $g$ in the little group of $\mathbf{k}_h$, from the overlap integrals:

$$x_{n,\mathbf{k}_h}(g) = \left\langle \mathbf{E}_{n,\mathbf{k}_h} | g\mathbf{D}_{n,\mathbf{k}_h} \right\rangle_{\text{UC}} \tag{10}$$

where $\mathbf{D} = \varepsilon\mathbf{E}$ is the displacement field and $\varepsilon$ the dielectric constant. From Schur's Orthogonality Relations[67], we can extract the symmetry vector $\mathbf{v}^T$ that gives the multiplicity of each irreducible representation (irrep) in the little group of each high-symmetry point. We label the

irrep according to the notation of Bilbao Crystallographic Server (BCS)[68].

This analysis returns, for the six lowest-electromagnetic modes:

$$\mathbf{v}^T = [(\blacksquare)^{2T} + \Gamma_2^- + \Gamma_4^-, R_4^- + R_5^+,$$
$$M_1 + 2M_4, X_1 + X_3 + X_4] \tag{11}$$

where $(\blacksquare)^{2T}$ indicates the irregular symmetry content at $\Gamma$ and $\omega = 0$ arising from transversality of the electromagnetic waves[42,43], with $()^T$ labeling the transverse bands.

A symmetry-constrained, tight-binding Hamiltonian $H(\mathbf{k})$ can be constructed for these transverse photonic bands, via the TETB methods

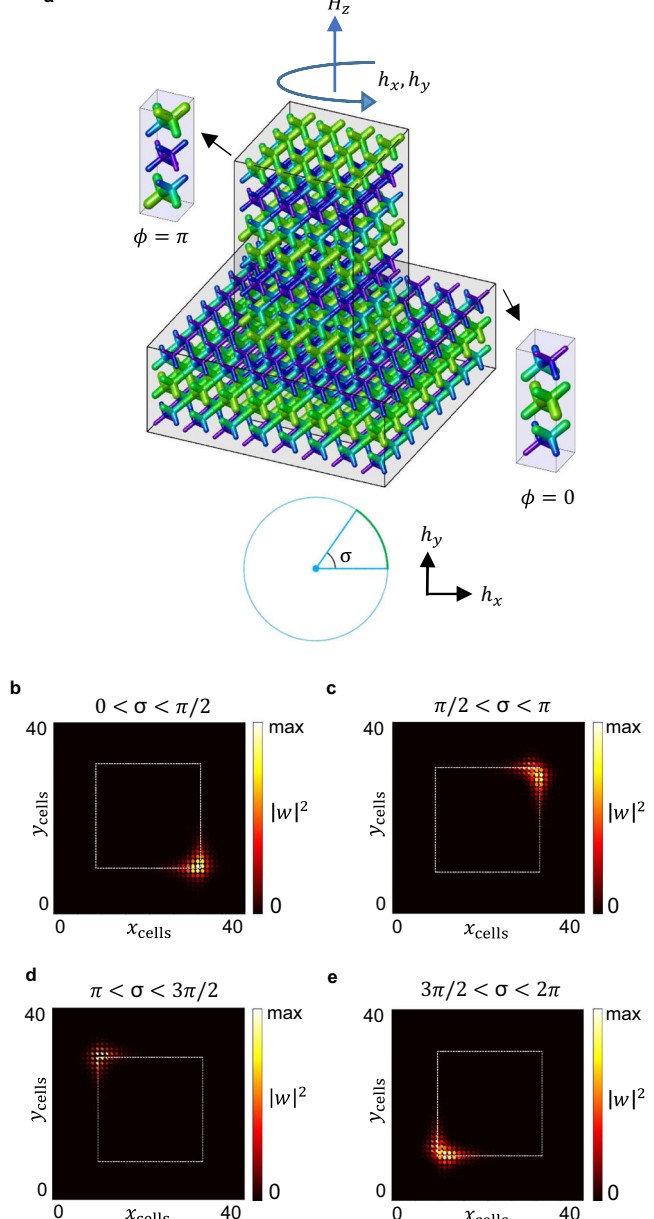

**a**

$H_z$

$h_x, h_y$

$\phi = \pi$

$\phi = 0$

$\sigma$

$h_y$

$h_x$

**b**

$0 < \sigma < \pi/2$

max

$|w|^2$

$y_{\text{cells}}$

$x_{\text{cells}}$

**c**

$\pi/2 < \sigma < \pi$

max

$|w|^2$

$y_{\text{cells}}$

$x_{\text{cells}}$

**d**

$\pi < \sigma < 3\pi/2$

max

$|w|^2$

$y_{\text{cells}}$

$x_{\text{cells}}$

**e**

$3\pi/2 < \sigma < 2\pi$

max

$|w|^2$

$y_{\text{cells}}$

$x_{\text{cells}}$

**Fig. 6 | Tunable AXI hinge states at Γ, for different magnetic bias configurations, computed via the TETB. a** displays the corresponding PhC dielectric structure. For visual purposes, the central $\phi = 0$ core is extruded vertically with respect to the phase-obstructed embedding with $\phi = \pi$. **b–e** correspond to α, γ, β, δ configurations. A single eigenvector is plotted here, upwards moving. The activation of the 90°-rotated hinges is made possible via a $h_{x,y}$ in-plane small bias component.

proposed by ref. 42. This approach involves the introduction of auxiliary longitudinal modes $\mathbf{v}^L$, which can regularize the Γ-point obstruction, such that $\mathbf{v}^{T+L} = \mathbf{v}^T + \mathbf{v}^L$ is regular. By exploiting a formal mapping between the Schrödinger and electromagnetic wave equations, which relates energies and frequencies quadratically ($\lambda \sim \omega^2$, see ref. 69), a TETB model is developed enforcing the lowest set of longitudinal bands at $\omega^2 \leq 0$, resulting in the $\mathbf{v}^T = \mathbf{v}^{T+L} - \mathbf{v}^L$ transverse vector capturing all the symmetry, topology and energetic features of the active bands in the PhC. For the specific $\mathbf{v}^T$ of Eq. (11), this can be achieved via:

$$\mathbf{v}^{T+L} = A_{2u}@4b + A_{2u}@4c. \tag{12}$$

with $\mathbf{v}^L = A_1@2a$, where the decomposition is done in terms of Elementary Band Representations (EBRs), which constitute the trivial atomic limits induced by a localized orbital at a specific Wyckoff position, as in the notation of BCS. This results in a 8-band model, from $A_{2u}$ photonic pseudo-orbitals at Wyckoff position $4b$ and $4c$. Gyrotropy can be as well modeled via non-minimal coupling to an external magnetic field **H**:

$$H(\mathbf{k}, \mathbf{H}) = H(\mathbf{k}) + f(\mathbf{k}, \mathbf{H}), \tag{13}$$

where the function $f(\mathbf{k}, \mathbf{H})$ should respect the symmetries of the crystal, **H** transforming as a pseudovector. Non-minimal coupling is adopted, due to the uncharged nature of photons, which prevents the use of Peierls substitution. The $\mathbf{H} = (0, 0, H_z)$ field is tuned in order for a Weyl dipole to form along the $k_z$ line, with a separation of $|\mathbf{Q}| = |\mathbf{q}_+ - \mathbf{q}_-| = 2\pi/N$ and $N \in \mathbb{N}$ and $N \geq 2$, as shown in Fig. 1a.

Starting from the $H(\mathbf{k}, \mathbf{H})$ magnetic Hamiltonian, we consider an additional perturbation aimed at capturing the effect of a SM of the dielectric elements. The perturbation is introduced as a $z$-periodic on-site potential of Wyckoff positions $4b$ and $4c$:

$$H_\Delta(\mathbf{r}, \mathbf{H}) = H(\mathbf{r}, \mathbf{H}) + \sum_i V_i \cos\left(\frac{2\pi z_i}{Na} + \phi\right) c_i^\dagger(\mathbf{r}) c_i(\mathbf{r}), \tag{14}$$

where $H(\mathbf{r}, \mathbf{H})$ is the real-space TETB Hamiltonian for the magnetic system before modulation, $a$ is the lattice parameter of the crystal before SM, and $V_i$ and $\phi$ parameterize the amplitude and the phase of the modulation, respectively. Note that the sum in Eq. (14) runs over all the basis pseudo-orbitals used in the TETB model, i.e., the pseudo-orbitals placed at WPs $4c$ and $4b$. Since these positions are related by symmetry, the amplitude of the modulation in the positions inside a WP should be equal. We will call them $V_{4c}$ and $V_{4b}$, respectively.

However, since the maximal Wyckoff position $4b$ and $4c$ cannot be adiabatically deformed into each other without breaking the symmetry of the model, we have the additional freedom of choosing the relative sign of their modulation amplitude, $V_{4b}$ and $V_{4c}$. Justified by the fact that the Wyckoff positions $4c$ fall inside the dielectric elements, while the Wyckoff positions $4b$ are in the air region, we decide to adopt the convention where the on-site potentials on $4b$ and $4c$ are opposite in sign, i.e., $V_{4c} = -V_{4b} > 0$, consistent with regions of higher and lower electromagnetic energy concentration. As shown in Fig. 1c, the effect of the SM is correctly captured by the transverse modes of the TETB after the introduction of the on-site potential, which results in the opening of a $C_z = 1$ gap.

## TETB symmetry vectors, double-band inversion and symmetry-constrained Γ-content

As we will demonstrate now, the supercell-modulated pseudo-orbitals of the TETB induce all the irreps of the supercell-modulated PhC band-structure, representing an exact representation for the $\tilde{\mathbf{v}}_\phi^T$ electromagnetic modes bellow the gap, ~ standing for the symmetry vector after modulation. Note that we express the symmetry vector in the notation of MSG #2.4 ($P\bar{1}$), which is the symmetry of the crystal after the introduction of the $H_z$ magnetic bias, the $\mathcal{I}$-symmetric SM and the off-axis $h_{x,y}$ perturbation. For the geometry-modulated PhCs, we find:

$$\begin{aligned}
\tilde{\mathbf{v}}_{\phi=0}^T = [(\blacksquare)^{2T} &+ 2\Gamma_1^+ + 2\Gamma_1^-, 2R_1^+ + 4R_1^-, \\
&3T_1^+ + 3T_1^-, 3U_1^+ + 3U_1^-, 2V_1^+ + 4V_1^-, \\
&3X_1^+ + 3X_1^-, 3Y_1^+ + 3Y_1^-, 3Z_1^+ + 3Z_1^-]
\end{aligned} \tag{15}$$

and

$$\tilde{\mathbf{v}}^T_{\phi=\pi} = [(\blacksquare)^{2T} + 2\Gamma_1^+ + 2\Gamma_1^- , 2R_1^+ + 4R_1^- ,$$
$$3T_1^+ + 3T_1^- , 3U_1^+ + 3U_1^- , 4V_1^+ + 2V_1^- , \quad (16)$$
$$3X_1^+ + 3X_1^- , 3Y_1^+ + 3Y_1^- , 3Z_1^+ + 3Z_1^-]$$

On the other hand, for the onsite-modulated TETB, we obtain:

$$\tilde{\mathbf{v}}^{T+L}_{\phi=0} = [4\Gamma_1^+ + 8\Gamma_1^- , 5R_1^+ + 7R_1^- ,$$
$$6T_1^+ + 6T_1^- , 6U_1^+ + 6U_1^- , 5V_1^+ + 7V_1^- , \quad (17)$$
$$6X_1^+ + 6X_1^- , 6Y_1^+ + 6Y_1^- , 6Z_1^+ + 6Z_1^-]$$

and

$$\tilde{\mathbf{v}}^{T+L}_{\phi=\pi} = [4\Gamma_1^+ + 8\Gamma_1^- , 5R_1^+ + 7R_1^- ,$$
$$6T_1^+ + 6T_1^- , 6U_1^+ + 6U_1^- , 7V_1^+ + 5V_1^- , \quad (18)$$
$$6X_1^+ + 6X_1^- , 6Y_1^+ + 6Y_1^- , 6Z_1^+ + 6Z_1^-]$$

The TETB therefore correctly models the double band inversion occurring between the system with $\phi = 0$ and $\phi = \pi$.

Table 1 shows the parity eigenvalues at the HSPs, with the HSP labeling done according to Bilbao Crystallographic Server (BCS) via the MKEV tool[52,53]. A double exchange of parity eigenvalues is observed at the $V = (\pi, \pi, 0)$ point.

After having identified the irregular irrep content at $\Gamma$, as $(\blacksquare)^{2T} = -\Gamma_1^+ + 3\Gamma_1^-$, consistent with symmetry-constrained decomposition for point group $\bar{1}$ as in refs. [42,43], we can split the TETB as follows: $\tilde{\mathbf{v}}^{T+L}_\phi = \tilde{\mathbf{v}}^T_\phi + \tilde{\mathbf{v}}^L_\phi$, where:

$$\tilde{\mathbf{v}}^L_{\phi=0,\pi} = [3\Gamma_1^+ + 3\Gamma_1^- , 3R_1^+ + 3R_1^- ,$$
$$3T_1^+ + 3T_1^- , 3U_1^+ + 3U_1^- , 3V_1^+ + 3V_1^- , \quad (19)$$
$$3X_1^+ + 3X_1^- , 3Y_1^+ + 3Y_1^- , 3Z_1^+ + 3Z_1^-]$$

represents the longitudinal auxiliary modes with $\omega^2 < 0$, and has the same expression for both $\phi = 0, \pi$. This shows that the symmetry vector of the TETB represents a precise representation of the electromagnetic modes below the gap of the rAXI. Specifically, the TETB symmetry vector with an onsite supercell-modulation can be decomposed as a longitudinal component $\tilde{\mathbf{v}}^L_\phi$ which has same expression for both $\phi = 0, \pi$ phases and a transverse part $\tilde{\mathbf{v}}^T_\phi$, which coincides with symmetry vector of the transverse modes of the PhC. As we will verify in the next section, $\tilde{\mathbf{v}}^L_\phi$ has trivial SI, so that the SI of the TETB model and of the MPB solutions coincide.

**Table 1 | Comparison of the parity eigenvalues for the two different axionic phases ($\phi = 0, \pi$)**

| HSP (h) | $\phi = 0$ | | $\phi = \pi$ | |
|---|---|---|---|---|
| Label (MSG #2.4) | $n_h^+$ | $n_h^-$ | $n_h^+$ | $n_h^-$ |
| $\Gamma$ | 4 | 8 | 4 | 8 |
| R | 5 | 7 | 5 | 7 |
| T | 6 | 6 | 6 | 6 |
| U | 6 | 6 | 6 | 6 |
| V | $\underline{5}$ | $\underline{7}$ | $\underline{7}$ | $\underline{5}$ |
| X | 6 | 6 | 6 | 6 |
| Y | 6 | 6 | 6 | 6 |
| Z | 6 | 6 | 6 | 6 |

$n_h^\pm$ count the number of modes with $\pm$ parity eigenvalues at the $\mathbf{k}_h$ HSP. Underline marks highlight the double exchange of the parity eigenvalues happening at the $V$ point.

## Magnetic symmetry-indicators for the $\mathcal{I}$-invariant photonic rAXI

In order to assess the role of $\mathcal{I}$-symmetry in the quantization of a relative axion angle, we apply the methods of Topological Quantum Chemistry (TQC) for non-fermionic systems[42,43,56,70–72] and analyze the symmetry indicators of the modulated PhCs, as the SM angle $\phi$ is varied. In particular, we consider the structures with $\phi = 0$ and $\phi = \pi$, in presence of both $H_z$ and a small in-plane $h_{x,y}$ which reduce the symmetry to MSG 2.4 (in the BNS notation of refs. [73,74]), and we evaluate the $\{\bar{z}_{2,x}, \bar{z}_{2,y}, \bar{z}_{2,z} | \bar{z}_4\}$ magnetic SI[18,50–53].

For the effective photonic TETB, which is regular and does not present any obstruction at $\Gamma$, the calculation of the SI follows directly from the well-known closed-formula expression that relates the $\mathcal{I}$-eigenvalues to the $\{\bar{z}_{2,x}, \bar{z}_{2,y}, \bar{z}_{2,z} | \bar{z}_4\}$ magnetic SI[18,50–53], i.e.:

$$\bar{z}_{2,i} = \frac{1}{2} \sum_{\substack{\mathbf{k}_h \in \{IIMS\} \\ \mathbf{k}_h \cdot \mathbf{R}_i = \pi}} (n_h^+ - n_h^-) \bmod 2 \quad (20)$$

$$\bar{z}_4 = \frac{1}{2} \sum_{\mathbf{k}_h \in \{IIMS\}} (n_h^+ - n_h^-) \bmod 4 \quad (21)$$

where $n_h^+$ ($n_h^-$) are the multiplicities of the positive (negative) parity eigenvalues at the high-symmetry point $\mathbf{k}_h$, and $\mathbf{R}_i$ are the primitive lattice vectors. This returns, depending on the phase $\phi$:

$$\nu^{T+L}_{\phi=0} = \{0, 0, 1 | 0\} \quad (22)$$

and

$$\nu^{T+L}_{\phi=\pi} = \{0, 0, 1 | 2\}. \quad (23)$$

To obtain the corresponding $\nu^T_\phi$ transverse SI for the electromagnetic modes, we can exploit the linearity of the SI with respect to the symmetry vector[43,75], i.e.:

$$\nu^T_i = \nu^{L+T}_i - \nu^L_i. \quad (24)$$

Since the SI of the longitudinal modes of Eq. (19) are trivial, it follows that the SI of the TETB and the MPB calculations coincide, $\nu^T_i = \nu^{L+T}_i$. This confirms that the $\phi = 0$ and the $\phi = \pi$ systems are obstructed with respect to each other, with a $\delta\bar{z}_4 = 2$ discontinuity of the *even* $\bar{z}_4$ signaling relative axion topology.

## Data availability

The crystal structure and dielectric parameters data for the axion insulators generated in this study are provided in the Source Data file. These data enable the replication of the full-wave electromagnetic results using MIT Photonic Bands Software[49]. Source data are provided with this paper.

## Code availability

Source codes for the computation of the bulk topological invariants of the 3D axion insulators are available from the corresponding authors upon request: a detailed explanation of the topological characterization for 3D photonic crystals will appear in an upcoming guided Tutorial publication, providing necessary generalizations from scalar[76] to vector fields[77].

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

## Acknowledgements
We wish to acknowledge recent discussions with Thomas Christensen from DTU about the origin-dependence of the $\bar{z}_4$ symmetry indicators and the role of unit cell shifts. A.G.E., M.G.V., A.M.P., M.G.D., I.R., and C.D. acknowledge support from the Spanish Ministerio de Ciencia e Innovación. A.G.E., A.M.P., and C.D. acknowledge support from the Gipuzkoa Provincial Council within the QUAN-000021-01 project, as well as the Basque Government Elkartek program (KK-2023/00016). A.G.E and M.G.V. acknowledge funding from the IKUR Strategy under the collaboration agreement between Ikerbasque Foundation and DIPC on behalf of the Department of Education of the Basque Government, Programa de ayudas de apoyo a los agentes de la Red Vasca de Ciencia, Tecnología e Innovación acreditados en la categoría de Centros de Investigación Básica y de Excelencia (Programa BERC) from the Departamento de Universidades e Investigación del Gobierno Vasco and Centros Severo Ochoa AEI/CEX2018-000867-S from the Spanish Ministerio de Ciencia e Innovación. M.G.V. thanks support to the Deutsche Forschungsgemeinschaft (DFG, German Research Foundation) GA 3314/1-1 - FOR 5249 (QUAST) and partial support from European Research Council (ERC) grant agreement no. 101020833. The work of J.L.M. has been supported in part by the Basque Government Grant No. IT1628-22 and the PID2021-123703NB-C21 grant funded by MCIN/AEI/10.13039/501100011033/ and by ERDF; "A way of making Europe". The work of B.B. and Y.H. is supported by the Air Force Office of Scientific Research under award number FA9550-21-1-0131. Y.H. received additional support from the US Office of Naval Research (ONR) Multidisciplinary University Research Initiative (MURI) grant N00014-20-1-2325 on Robust Photonic Materials with High-Order Topological Protection. C.D. acknowledges financial support from the MICIU through the FPI PhD Fellowship CEX2018-000867-S-19-1. M.G.D. acknowledges financial support from the Government of the Basque Country through the pre-doctoral fellowship PRE_2022_2_0044.

## Author contributions
A.G.E., M.G.V., C.D. and M.G.D. initiated the project. A.G.E., M.G.V., B.B. and J.L.M. outlined the work. A.G.E., M.G.V., C.D., A.M.P., I.R., Y.H. and B.B. developed the theory. C.D., A.M.P., A.G.E. and M.G.V. performed the simulations. All the authors discussed and analyzed the results. C.D., A.M.P., M.G.V., A.G.E. and Y.H. wrote the manuscript with input from all coauthors. A.G.E. and M.G.V. coordinated the project.

## Funding

## Competing interests
The authors declare no competing interests.
