## [Peer review file · Nature Communications]

Axion Topology in Photonic Crystal Domain WallsREVIEWER COMMENTS

Reviewer #1 (Remarks to the Author):

In the present work, entitled “Axion Topology in Photonic Crystal Domain Walls”, C. Devescovi and colleagues, propose a theoretical design of a photonic crystal containing a gyrotropic material leading to axion-like response and topology. The proposed system can be realized experimentally and may offer unidirectional topological hinge states. One of the advantages of the system emphasized by the authors is tunability by magnetic field. Overall, the paper is well-written, and results are scientifically sound.

My main concern with this work is related to the lack of breakthrough novelty. I understand that the authors may have a non-electromagnetic/photonic background and may not be aware that the term they refer to as axion term, leading to the magneto-electric coupling, is well-known in photonic as bi-anisotropic response. As such, this response has been studied quite intensely in the context of topological photonic crystals. Moreover, there is a huge variety of responses that can be engineered, both breaking time-reversal symmetry (e.g., Tellegen-like response) and not (e.g., chiral response). In fact, the first symmetry protected topological photonic phases proposed was based on magneto-electric coupling of this kind [1], experimentally realized [2,3], and even extended to 3D [4]. Topological systems with magneto-electric coupling due to ferrites was also proposed in Ref. [5], and gained more attention in recent studies [6,7]. Thus, I do not see how this work, while presenting an original result, can be suitable for Nature Communications. I think impact falls a bit short. I would definitely have a different stance if the work was accompanied by experimental study, as no axion/Tellegen topological system has been presented experimentally to the best of my knowledge.

To summarize, while I find this work interesting and presenting an original results on topology a 3D system with engineered axion-like response, I do not find it groundbreaking and original enough to appear in Nature Communications. I believe other journal, such as Physical Review journals or Scientific report, would be more appropriate for the dissemination. This is especially true since one-way hinge states has been already proposed in a simpler system based on an electric circuit [8], which is much more easy to reconfigure and tune, which the authors present as one of the major selling points of their design.

[1] A. B. Khanikaev, et al., Nature materials 12 (3), 233-239 (2012).

[2] X. Cheng, et al., Nature materials 15 (5), 542-548 (2016).

[3] K. Lai, et al., Scientific reports 6 (1), 28453 (2016).

[4] A. Slobozhanyuk, et al. Three-dimensional all-dielectric photonic topological insulator
Nature Photonics 11 (2), 130-136 (2017).

[5] DA Jacobs, et al., New Journal of Physics 17 (12), 125015 (2015).

[6] Filipa R. Prudêncio and Mário G. Silveirinha, Phys. Rev. Applied 19, 024031 (2023).

[7] J. C. Serra and M. G. Silveirinha, arXiv:2309.15320 (2023).

[8] X. Ni, et al., Phys. Rev. Applied 13, 064031 (2020).

Reviewer #2 (Remarks to the Author):

In the manuscript entitled “Axion Topology in Photonic Crystal Domain Walls”, the authors present a theoretical study of a photonic axion insulator in a 3D magnetic photonic crystal that sustains tunable chiral hinge states along the domain walls. In the theoretical analysis, they employ a transversality-enforced Tight-binding (TETB) model and calculate the magnetic symmetry indicators by analyzing the parity eigenvalue at the high-symmetry points. In the numerical simulations, they use a 3D magnetic photonic crystal with tunable magnetization direction and calculate the slab Wilson loops by counting the winding of the Berry phase. As far as I know, this paper proposes and demonstrates the first theoretical model and general design strategy of photonic axion insulators in 3D magnetic photonic crystals. Therefore, it is definitely interesting for a wide audience in the research field of topological photonics. Before I can recommend its publication in Nature Communications, some questions/comments need to be clarified and addressed.

1, The authors use both the magnetic symmetry indicators and the slab Wilson loops to verify the topology of the axion insulator, what are the differences? It would be better if the authors could provide a parity eigenvalue at high-symmetry points of the magnetic photonic crystals to understand the magnetic symmetry indicators.

2, Since the chiral hinge states in the manuscript are similar to the topological one-way fiber [Nat. Commun. 9, 5384 (2018)], can the authors comment on their differences and highlight this point in the manuscript?

3, Could the authors provide a reference that discusses the details of the permittivity tensor when the direction of magnetization is changed?

4, Since the prominent topological feature of the photonic axion insulators is the unidirectional and robust propagation of light, can the authors conduct full wave simulations to demonstrate the one-way and defect-immune propagation of the chiral hinge states?

5, In the present manuscript, the photonic chiral hinge states only emerge in the domain wall between two magnetic photonic crystals, can the chiral hinge states exist on the outer hinges of a single photonic crystal? If yes, can the author provide a design strategy?

Reviewer #3 (Remarks to the Author):

The article "Axion Topology in Photonic Crystal Domain Walls" by C. Devescovi et al. proposes a general strategy to design axion insulators in gyrotropic photonic crystals. As far as the reviewer knows, this is the first time that such a topological phase has been found in any photonic system. The design strategy is based on the introduction of a static, geometric supercell modulation of the lattice dielectric elements.

The obtained axionic modes of light propagate in the form of unidirectional and non-reciprocal channels localized at the hinges of the crystal. Such modes could be used in guided-light communication networks. Moreover, the article proposes a way of manipulating the axionic channels through a small external magnetic bias. Such a proposal could be used as an effective switch to guide light between various one-dimensional light channels, controlling the direction of the flow of light.

Besides, intriguingly, axion insulators couple with electromagnetism in the bulk similarly to dark matter axion-like particles. Based on this similarity, recent studies have suggested that axion insulators, as the one presented in this article, could be used as axion dark-matter particle detectors.

Overall, the manuscript is very interesting work. The theory and analysis are solid and well supported by the data and discussion. Without experiments, it will lead many interesting works in topological photonics and the closed related field. It is still worthwhile to be considered in Nature Communications.

The detailed questions and comments are following:

1) How feasible would it be to build such axion insulators experimentally?

2) The article uses a recently developed method to build tight binding models of 3D photonic crystals. The intrinsic vectorial nature of electromagnetic field solutions in 3D Photonic crystals have precluded

the development of such models in the past. How easily could one adopt this strategy to develop models of crystals in any space group with an arbitrary unit cell?

3) The following references could be added to reflect the state-of-the-art.

<https://www.nature.com/articles/s41467-022-30909-0>

<https://www.sciencedirect.com/science/article/abs/pii/S0010465522002120>

<https://journals.aps.org/prb/abstract/10.1103/PhysRevB.99.235423>

<https://journals.aps.org/prb/abstract/10.1103/PhysRevB.101.195105>

<https://onlinelibrary.wiley.com/doi/full/10.1002/adom.201900900>

<https://www.degruyter.com/document/doi/10.1515/nanoph-2019-0451/html>

Axion Topology in Photonic Crystal Domain Walls: Point-by-point response to the reviewers' comments

Chiara Devescovi^{*,1,*} Antonio Morales-Pérez^{*,1} Yoonseok Hwang,² Mikel García-Díez,^{1,3} Iñigo Robredo,^{4,1} Juan Luis Mañes,³ Barry Bradlyn,² Aitzol García-Etxarri,^{1,†} and Maia G. Vergniory^{4,1,‡}

¹*Donostia International Physics Center, Paseo Manuel de Lardizabal 4, 20018 Donostia-San Sebastian, Spain.*

²*Department of Physics, University of Illinois at Urbana-Champaign, Urbana, IL, USA*

³*Physics Department, University of the Basque Country (UPV/EHU), Bilbao, Spain*

⁴*Max Planck Institute for Chemical Physics of Solids, Dresden D-01187, Germany*

(Dated: May 3, 2024)

We thank the Referees for the time devoted to reviewing our manuscript: We believe that their input has allowed us to substantially improve the quality of our manuscript. In the following, we respond point by point to all their questions and concerns, indicating the changes we implemented in our article accordingly. Our point-by-point responses are written in black and provided with the original comments reproduced verbatim in blue.

REFEREE NO. 1

In the present work, entitled 'Axion Topology in Photonic Crystal Domain Walls', C. Devescovi and colleagues, propose a theoretical design of a photonic crystal containing a gyrotropic material leading to axion-like response and topology. The proposed system can be realized experimentally and may offer unidirectional topological hinge states. One of the advantages of the system emphasized by the authors is tunability by magnetic field. Overall, the paper is well-written, and results are scientifically sound. My main concern with this work is related to the lack of breakthrough novelty. I understand that the authors may have a non-electromagnetic/photonic background and may not be aware that the term they refer to as axion term, leading to the magneto-electric coupling, is well-known in photonic as bi-anisotropic response.

We thank the referee for reviewing our work and for their comments. Addressing first the referee's concern about the electromagnetic/photonic background of the authors, we would like to highlight that one of the corresponding authors of this work, Aitzol Garcia-Etxarri, has more than 15 years of experience in research on photonics. In particular, one of his main research areas is focused on the optical response of chiral/bi-anisotropic and dual dielectric systems. As examples of his expertise, we include a selection of his articles on related topics here [1–14]. As a result, the relation between magneto-electric coupling and bi-anisotropy is well familiar to the group of authors.

For the sake of clarity, we would like to point out a very common misconception. When talking about axions, it is very important to differentiate between axionic response and axion topology. *Axion response* refers to a magnetoelectric coupling in Maxwell's equations which emerges in bi-anisotropic media and it is related with Tellegen media. *Axion topology*, instead, is a property of a band-structure that leads to the emergence of topological chiral hinge modes.

In electronic crystals, axion topology implies an axion response. This connection traces back to how electric fields couples to charge particles [15–19]. However, our manuscript deals with photonic crystals. Hence, our systems are not electrically charged, forbidding this direct connection between axion topology and magnetoelectric coupling. It is worth mentioning that the existence (or not) of such a connection constitutes a fascinating open problem that deserves to be studied further in the future. Despite the absence of this connection, axion topology leads to the emergence of topologically protected chiral hinge modes, not only in electronic systems but also in photonic systems, as confirmed by the present study.

Moreover, and most importantly, in our design, axion topology arises exclusively due to the topological properties encoded in the photonic bands. As we will highlight several times in this reply, we do not employ any kind of bi-anisotropy in our system. The axion topology of our system relies exclusively the gyrotropic nature and periodicity of the crystal. We found the referee's comments particularly stimulating. Since we believe that other readers may have similar questions, in the new version of the manuscript, we address the relation between bi-anisotropy, Tellegen media, and axion response in the new section "**Bi-anisotropic media and Tellegen responses.**" Moreover, we also analyze briefly the connection between axion insulators and magneto-electric coupling in electronic systems in the new "**Magneto-electric response of condensed matter axion insulators**" section of the manuscript.

As such, this response has been studied quite intensely in the context of topological photonic crystals. Moreover, there is a huge variety of responses that can be engineered, both breaking time-reversal symmetry (e.g., Tellegen-like repone) and not (e.g., chiral response). In fact, the first symmetry protected topological photonic phases proposed was based on magneto-electric coupling of this kind [1], experimentally realized [2,3], and even extended to 3D [4]. Topological systems with magneto-electric coupling due to ferrites was also proposed in Ref. [5], and gained more attention in recent studies [6,7].

[1] A. B. Khanikaev, et al., Nature materials 12 (3), 233-239 (2012).

[2] X. Cheng, et al., Nature materials 15 (5), 542-548 (2016).

[3] K. Lai, et al., Scientific reports 6 (1), 28453 (2016).

[4] A. Slobozhanyuk, et al. Three-dimensional all-dielectric photonic topological insulator Nature Photonics 11 (2), 130-136 (2017).

[5] DA Jacobs, et al., New Journal of Physics 17 (12), 125015 (2015).

[6] Filipa R. Prudêncio and Mário G. Silveirinha, Phys. Rev. Applied 19, 024031 (2023).

[7] J. C. Serra and M. G. Silveirinha, arXiv:2309.15320 (2023).

The works [20–24] cited by the referee predominantly focus on the use of bi-anisotropy to induce and protect topological phases. For instance, Reference [20] demonstrates using material bi-anisotropy to mimic spin-orbit coupling in light to emulate photonic analogs of the quantum spin Hall effect in 2D. Other of the cited references, like [23], use bianisotropy engineered in the sub-components at the unit cell level to open a \mathbb{Z}_2 gap in a 3D Dirac semimetal. These studies show that bi-anisotropy can effectively induce topological effects in photonic band structures. Nevertheless, our contribution strongly contrasts with the works cited in that we obtain a novel topological phase in a system where bi-anisotropy is inherently absent both in the meta-atom design and in the dielectric medium.

Our approach further contrasts with the other studies referenced by the referee [25, 26], which propose to employ time-dependent dynamical driving to induce an axionic topological phase. In our structure, axion topology arises solely from the periodicity of a static photonic lattice and inversion symmetry quantization.

Thus, I do not see how this work, while presenting an original result, can be suitable for Nature Communications. I think impact falls a bit short. I would definitely have a different stance if the work was accompanied by experimental study, as no axion/Tellegen topological system has been presented experimentally to the best of my knowledge.

Addressing the referee's concern about the lack of experimental evidence, taking into account that we agree with them that an experimental realization could be developed, we have introduced new sections in our manuscript detailing practical steps toward realizing the axion-insulating phase in photonic crystals. Although our study remains theoretical and supported by comprehensive electromagnetic simulations, the sections "**Key Elements of Relative Axion Design**" and "**Suitable experimental platforms**" outline experimental strategies and identify existing platforms suitable for our design. We also stress that our photonic axion proposal does not require bi-anisotropic media and solely employs gyrotropic periodic materials. Please note that very strong gyrotropic responses can be effectively realized within state-of-the-art techniques in photonic crystals at microwave frequencies [27], highlighting the feasibility of our proposed implementation.

To summarize, while I find this work interesting and presenting an original results on topology a 3D system with engineered axion-like response, I do not find it groundbreaking and original enough to appear in Nature Communications. I believe other journal, such as Physical Review journals or Scientific report, would be more appropriate for the dissemination. This is especially true since one-way hinge states has been already proposed in a simpler system based on an electric circuit [8], which is much more easy to reconfigure and tune, which the authors present as one of

the major selling points of their design. [8] X. Ni, et al., *Phys. Rev. Applied* 13, 064031 (2020).

We appreciate the referee's comparison of our tunable axionic hinge states with the tunable one-way hinge states found in electric circuits. However, we wish to clarify that the hinge states in our setup emerge *as a consequence of* the axion topology we investigate. This key distinction sets our work apart from the referenced electrical circuit-based systems, which, unlike our photonic platform, do not exhibit any axion topology nor enable magneto-optic control. Far from being a mere proposal of tunable hinge states, our work introduces a topological photonic phase that, to our knowledge, has not been previously explored within 3D photonic crystals. This phase, inspired by condensed matter physics - where it remains elusive and challenging to observe - shows significant promise for photonic implementations. The first reason is that the mature fabrication techniques in 3D gyrotropic photonic crystals hint at possible realization in gyromagnetic setups. The second reason is that the topological term in axion insulators directly relates to the axion-photon coupling mechanism. In the present study, we've demonstrated the application of axion topology for inducing topological switching and control of magneto-optical photonic axionic channels. Looking ahead, the scope of our photonic design aims to further the exploration of axion dark matter to bridge traditional and topological approaches for axionic matter detection. All these capabilities would not be achievable in electrical circuits-based systems. Thus, we believe that our work's originality and potential impact align well with the scope of Nature Communications, offering significant contributions to the field of 3D topological photonics. In response to the referee's concern, we've adjusted our manuscript to focus more on the significance of axion topology in a photonic setup than on the simple reconfigurability or tunability of its axionic hinge states. The new version highlights these new comments and sentences.

REFEREE NO. 2

In the manuscript entitled “Axion Topology in Photonic Crystal Domain Walls”, the authors present a theoretical study of a photonic axion insulator in a 3D magnetic photonic crystal that sustains tunable chiral hinge states along the domain walls. In the theoretical analysis, they employ a transversality-enforced Tight-binding (TETB) model and calculate the magnetic symmetry indicators by analyzing the parity eigenvalue at the high-symmetry points. In the numerical simulations, they use a 3D magnetic photonic crystal with tunable magnetization direction and calculate the slab Wilson loops by counting the winding of the Berry phase. As far as I know, this paper proposes and demonstrates the first theoretical model and general design strategy of photonic axion insulators in 3D magnetic photonic crystals. Therefore, it is definitely interesting for a wide audience in the research field of topological photonics. Before I can recommend its publication in Nature Communications, some questions/comments need to be clarified and addressed. 1. The authors use both the magnetic symmetry indicators and the slab Wilson loops to verify the topology of the axion insulator, what are the differences? It would be better if the authors could provide a parity eigenvalue at high-symmetry points of the magnetic photonic crystals to understand the magnetic symmetry indicators.

We thank the referee for raising this interesting question. Symmetry Indicators (SIs) and slab Wilson loops offer distinct approaches to characterizing topological phases in photonic crystals. SIs, rooted in Topological Quantum Chemistry, utilize symmetry arguments to ‘indicate’ and probe the topological nature of band structures, focusing on the sole symmetry content at High-Symmetry Points (HSPs), which is computationally a desirable property. However, their characterization can, in certain cases, be insufficient to fully resolve the topological properties. A clear example is the case of the $\bar{z}_{2,i}$ indicator in the inversion-symmetric magnetic space group #2.4. Though related to the parity of associated Chern number (C_i):

$$C_i = \bar{z}_{2,i} \pmod{2}, \quad (1)$$

the symmetry indicator alone cannot fully determine its value. Conversely, slab Chern numbers provide a more direct method to extract the topological invariants by evaluating the Wilson loop of a slab, allowing for the identification of half-quantized Hall conductivities and offering an unambiguous characterization of relative axion topology [28, 29], even in complex scenarios where SIs may falter, though at the cost of a computationally more demanding calculation. In our topological characterization of relative axion insulators, we employed both methods to achieve a comprehensive characterization of the topological phases within photonic crystals, specifically targeting the observable properties of axion topology. The two main reasons are the following. SIs, being computed locally in momentum space at the HSPs, might miss the effects of other features in the Brillouin zone, which can obscure the axion topology, such as the presence of other gapped bands overlapping spectrally with the relevant axion gap or the presence of Chern surface states, the latter being particularly relevant in our case. Moreover, the $\bar{z}_4 = 2$ symmetry indicator, despite being associated with axion topology, becomes origin-dependent in the presence of an anomalous Hall background contribution [28, 29], which again is exactly what happens in our case, due to the presence of a non-zero Chern vector in the bulk of the PhC. Because of these limitations, despite observing a $|\delta\bar{z}_4 = 2|$ discontinuity between the $\phi = 0, \pi$ bulk bands and a double parity inversion on the V -point, we also relied on slab Wilson loops for a conclusive characterization of the physically observable effects of *relative* axion topology. It’s crucial to highlight that beyond the primary focus on topological characterization, the explicit calculation of the \bar{z}_4 indicator underscores a clear connection between relative axion topology with the shift in the supercell modulation, a point probably not stress enough in our first submissstress the importance of the origin-dependence of magnetic symmetry indicators in enabling relativivion. To elaborate more on the origin-dependence of the \bar{z}_4 and its relation with the relative shift, we added two new supplementary subsections, named **"Origin-dependence of the \bar{z}_4 symmetry indicator"** and **"Engineering axion topology through unit cell shifts"**. As further suggested by the referee, to allow a better understanding of the characterization via magnetic SIs, we provide herewith the parity eigenvalues at the HSPs and show explicitly their relation with the SIs. The table I is incorporated into the Methods section of the manuscript. The parity eigenvalues are related to the magnetic SIs

HSP (h)	$\phi = 0$		$\phi = \pi$	
Label (MSG #2.4)	n_h^+	n_h^-	n_h^+	n_h^-
Γ	4	8	4	8
R	5	7	5	7
T	6	6	6	6
U	6	6	6	6
V	5	7	7	5
X	6	6	6	6
Y	6	6	6	6
Z	6	6	6	6

Table I: Comparison of the parity eigenvalues for the two different axionic phases ($\phi = 0, \pi$). n_h^\pm count the number of modes with \pm parity eigenvalues at the \mathbf{k}_h HSP. The HSP labeling is done according to Bilbao Crystallographic Server (BCS) via the MKEV tool [30, 31]. Underline marks highlight the double exchange of the parity eigenvalues happening at the V point.

($\{\bar{z}_{2,x}, \bar{z}_{2,y}, \bar{z}_{2,z} | \bar{z}_4\}$) via the closed-formula expressions:

$$\bar{z}_{2,i} = \frac{1}{2} \sum_{\substack{\mathbf{k}_h \in \{IIMs\} \\ \mathbf{k}_h \cdot \mathbf{R}_i = \pi}} (n_h^+ - n_h^-) \pmod{2} \quad (2)$$

$$\bar{z}_4 = \frac{1}{2} \sum_{\mathbf{k}_h \in \{IIMs\}} (n_h^+ - n_h^-) \pmod{4} \quad (3)$$

where n_h^+ (n_h^-) are the multiplicities of the positive (negative) parity eigenvalues at the \mathcal{I} -invariant high-symmetry point \mathbf{k}_h , and \mathbf{R}_i are the primitive lattice vectors. This returns, depending on the phase ϕ :

$$\nu_{\phi=0}^{T+L} = \{0, 0, 1|0\} \quad (4)$$

$$\nu_{\phi=\pi}^{T+L} = \{0, 0, 1|2\}. \quad (5)$$

2, Since the chiral hinge states in the manuscript are similar to the topological one-way fiber [Nat. Commun. 9, 5384 (2018)], can the authors comment on their differences and highlight this point in the manuscript?

The one-way fibers discussed in Ref. [32] and our hinge states share the characteristic of chiral unidirectional propagation, a feature also observed in other topological edge states within gyrotropic photonic environments due to the breaking of time-reversal symmetry. Despite these similarities, significant differences exist in their physical origins, potential applications, structural properties, and topological protection mechanisms.

The one-way guided modes from the referenced study arise from defects linked to screw dislocations within the crystal's periodic structure and are protected by a second-Chern number. In contrast, the hinge states we propose are higher-order topological insulator modes that originate from inversion-symmetry quantization and are protected by an axion invariant. This fundamental distinction in origin and protection mechanism leads to differing properties between the two types of states.

A practical distinction is our ability to manipulate the location of axion hinge states on any of the four hinges of the structure (Fig.6), unlike the one-way fibers, where localization is strictly determined by the existing dislocations.

Additionally, the proposed fabrication process for the two structures diverges significantly. Our method involves establishing a domain wall between two photonic crystals, whereas the one-way fibers require a continuously tailored pattern of screw dislocations overlaid on a pre-existing supercell modulation.

Another fundamental structural difference concerns the bending characteristics and the topological propagation. In an axion insulator, a finite structure in all three dimensions would exhibit a closed string wrapping around the central axion core [33], evidenced in our simulations by the presence of both an upward-moving and downward-moving state at inversion-related hinges (Figs. 5-6). This contrasts with the one-way modes propagation of the helix, which propagates unidirectionally along the vertical dislocation, and where, as the authors show, fiber bends can disrupt the 3D bandgap.

Overall, while both systems share the trait of unidirectional propagation, the topological properties, mechanisms of protection, possible fabrication and applications of our hinge states and the one-way fibers differ markedly. This highlights the specificity of our chiral hinge states.

3, Could the authors provide a reference that discusses the details of the permittivity tensor when the direction of magnetization is changed?

In gyrotropic media, the magnetically-induced anisotropy in the permittivity tensor is controlled by the gyration vector $\boldsymbol{\eta} = (\eta_x, \eta_y, \eta_z)$ according to the relation:

$$\boldsymbol{\varepsilon} = \begin{pmatrix} \varepsilon_{xx} & i\eta_z & -i\eta_y \\ -i\eta_z & \varepsilon_{yy} & i\eta_x \\ +i\eta_y & -i\eta_x & \varepsilon_{zz} \end{pmatrix}, \quad (6)$$

where $\boldsymbol{\eta} = \boldsymbol{\eta}(\mathbf{H}^{\text{ext}})$ depends by the applied external magnetic field \mathbf{H}^{ext} . In our case, we assumed the simplest scenario of an isotropic medium, for which:

$$\boldsymbol{\eta}(\mathbf{H}^{\text{ext}}) = c\mathbf{H}^{\text{ext}}, \quad (7)$$

with $c > 0$. Correspondingly, as the magnetic field is rotated along a given i direction, each of the η_i components in the permittivity gets activated. For reference, these relations are discussed in detail in textbooks such as [34] and also briefly covered in [35]. These references have been added to the main text.

4, Since the prominent topological feature of the photonic axion insulators is the unidirectional and robust propagation of light, can the authors conduct full wave simulations to demonstrate the one-way and defect-immune propagation of the chiral hinge states?

We thank the referee for this question and appreciate the opportunity to discuss the methodologies and computational strategies that have guided our study. Initially, we intended to perform full-wave electromagnetic simulations on the xy confined 3D rod geometries for the demonstration of the axion hinge states (Figs. 5-6) expected by our theoretical analysis. However, due to computational challenges, even within an HCP computing cluster, we encountered significant limitations, particularly the impossibility of simulating more than $16 \times 16 \times 3$ unit cells in a PhC without encountering dramatic finite-size effects. This challenge necessitated the development of an electromagnetic transversality-enforced tight-binding (TETB) model to enable the simulation of the supercells required for observing the axionic hinge states. As detailed in the Methods section, this TETB model is fully constrained by transversality and symmetry, adhering strictly to Maxwell's equations constraints and the crystalline symmetries of the dielectric distribution. In our Ref. [36], we demonstrated the TETB method to faithfully reproduce all the symmetry-indicated topologies, such as inversion axion topology, offering a robust alternative to direct full-wave simulations to test topological properties.

We believe the referee's suggestion also hints at applying time-domain simulations for dynamic analyses of chiral propagation and disorder effects in our structure, which would require a very different approach from the one employed here. Our present investigation, in the frequency domain, exploits topological band theory, focusing on momentum-space analysis, to elucidate the topological properties and boundary phenomena of the bands. Notably, the unidirectionality of the hinge states is evidenced by the presence of chiral channels that span the topological gap, without any counter-propagating modes, offering evidence of topological protection that's observable even without direct time-domain visualization.

It’s important to note that our simulations of surface states and domain walls (Fig. 4 of the main text and Fig. S1 of the Supplemental Materials) with axion topology were conducted using full-wave electromagnetic simulations in MPB, showcasing our commitment to leveraging direct full-wave electromagnetic simulation techniques where feasible.

Given the computational and methodological constraints mentioned, further exploration of the hinge states’ dynamic propagation characteristics, particularly through time-domain FDTD simulations, would necessitate a dedicated study beyond the current scope of our proposal. Additionally, the specific role of defects within the context of axion insulators has been explored in literature, indicating that quenched disorder, which maintains average inversion symmetry, does not lead to the delocalization of axionic states [37].

In summary, the complexities associated with direct full-wave simulations in our structures have eventually led us to develop alternative methodologies, such as TETB approaches, that allow us to probe the topological properties and observe the axionic hinge channels of our electromagnetic system, presenting a robust alternative to direct full-wave simulations suited for extended 3D PhC supercell geometries.

5, In the present manuscript, the photonic chiral hinge states only emerge in the domain wall between two magnetic photonic crystals, can the chiral hinge states exist on the outer hinges of a single photonic crystal? If yes, can the author provide a design strategy?

We thank the referee for highlighting the concept of an *intrinsic* axion insulator, which inherently differs from the *relative* axion insulator discussed in our manuscript. This allows us to underscore the unique aspects of our design and to discuss its potential advantages when contrasted with the *intrinsic* axion insulator scenario. An *intrinsic* axion insulator phase manifests independently within a single photonic crystal, without the necessity for a domain wall interface. Designing an *intrinsic* axion insulator phase extends beyond the scope of the current manuscript. However, key steps for such a design are outlined below:

1. **Integration of Magnetism:** Ensuring the presence of magnetism, through either gyrotropic or gyroelectric effects, is crucial for breaking time-reversal symmetry within the photonic crystal.
2. **Specific Crystal Symmetries:** The photonic crystal’s symmetries must simultaneously:
 - (a) Ensure all Chern numbers equal zero ($C_i = 0$).
 - (b) Quantize the axion magneto-electric angle ($\theta = \pi$).
 Symmetries meeting these criteria, such as $T \times \{1|(1, 1, c/2)\}$ —time-reversal combined with half-lattice translation—are discussed in Ref. [Physical Review B 86.11 (2012): 115112], based on crystallographic arguments.
3. **Opening of a Topological Bulk Gap:** Opening a magnetic bulk gap poses a significant challenge in 3D photonics. To address this task, in the present work, we follow a deterministic approach, via the creation and annihilation of Weyl points.
4. **Preventing radiation in the continuum:** For intrinsic axion insulators, it’s critical to ensure that the bulk topological bandgap is positioned outside the light cone continuum to prevent hinge states from radiating externally. Finding strategies to achieve this is not an easy task, and it is something that should be addressed very carefully.

Please note that in our design, inversion symmetry was able to quantize the axion angle (point 2.b) but could not prevent the existence of non-zero Chern numbers (point 2.a): therefore, in order to make the *relative* axion topology emerge, we had to get rid of the anomalous Hall contribution via construction of the domain wall. On the other hand, one of the advantages of our *relative* axion design is that the observed hinge states are automatically protected from radiation in the continuum (point 4) since the embedded rod geometry fully confines them.

REFEREE NO. 3

The article "Axion Topology in Photonic Crystal Domain Walls" by C. Devescovi et al. proposes a general strategy to design axion insulators in gyrotropic photonic crystals. As far as the reviewer knows, this is the first time that such a topological phase has been found in any photonic system. The design strategy is based on the introduction of a static, geometric supercell modulation of the lattice dielectric elements.

The obtained axionic modes of light propagate in the form of unidirectional and non-reciprocal channels localized at the hinges of the crystal. Such modes could be used in guided-light communication networks. Moreover, the article proposes a way of manipulating the axionic channels through a small external magnetic bias. Such a proposal could be used as an effective switch to guide light between various one-dimensional light channels, controlling the direction of the flow of light.

Besides, intriguingly, axion insulators couple with electromagnetism in the bulk similarly to dark matter axion-like particles. Based on this similarity, recent studies have suggested that axion insulators, as the one presented in this article, could be used as axion dark-matter particle detectors.

Overall, the manuscript is very interesting work. The theory and analysis are solid and well supported by the data and discussion. Without experiments, it will lead many interesting works in topological photonics and the closed related field. It is still worthwhile to be considered in Nature Communications.

The detailed questions and comments are following:

1) How feasible would it be to build such axion insulators experimentally?

We thank the referee for this question. We are optimistic about the feasibility of experimentally realizing relative axion insulator domain walls using state-of-the-art technologies in gyrotropic photonic crystals, particularly at microwave frequencies. Actually, we have identified a specific experimental setup that presents all the necessary requirements for implementing this design. The realization of the relative axion domain wall relies on a few critical requirements and design steps:

1. Design of a time-reversal symmetry (TRS) broken Weyl semimetal:

- (a) Weyl point annihilation: Fundamental for the emergence of axion-type electrodynamics, leading to a gapped 3D Chern phase with non-zero Chern vectors.
- (b) Inversion symmetry preservation: Crucial for quantizing the axion angle.

2. Creation of a Domain Wall Configuration:

- (a) Alignment of Chern vectors: Necessary to eliminate anomalous Hall surface states.
- (b) Relative shift of the inversion-symmetry center: Essential for manifesting the relative axion topology, by creating a shift along the Chern vector.

Among the designs fulfilling these criteria, the gyrotropic layered 3D Chern structure, proposed by Baile Zhang's group in Ref.[27], exemplifies a feasible implementation in microwaves. This design presents Chern vectors from Weyl point annihilation (1a), and inversion symmetry preservation (1b) for an extensive domain of the parameter space, as illustrated by the horizontal $R = 0$ line in Fig. 1b of the referenced work. As well, its domain walls can be oriented to present aligned and rotated Chern vectors (2a). The introduction of a relative inversion-symmetry shift along the z -axis (2b), along the c -axis of the crystals, could be achieved in the same setup. Please note that, since not relying on spin-orbit coupling, the design principle for our relative axion insulator is potentially transferable to other inversion-symmetric platforms that break TRS, beyond photonic crystals. For example, it could be implemented in acoustic metamaterials, where non-reciprocal circulating airflow can emulate non-reciprocal gauge fields and open Chern gaps [38–42]. For summarizing the key design requirements and the possible experimental platform suited for realizing relative axion topological nodal walls, in the main text, we have introduced two new sections: '**Key Elements of the Relative Axion Design**' and '**Possible experimental implementation of the relative AXIs**'.

2) The article uses a recently developed method to build tight-binding models of 3D photonic crystals. The intrinsic vectorial nature of electromagnetic field solutions in 3D Photonic crystals has precluded the development of such models in the past. How easily could one adopt this strategy to develop models of crystals in any space group with an arbitrary unit cell?

The transversality-enforced tight binding (TETB) method is versatile, and applicable across all 230 space groups (SGs) for photonic crystals with any arbitrary unit cell (UC). Additionally, it adapts to incorporate gyrotropy via non-minimal coupling, making it suitable for modeling certain magnetic space groups (MSGs). In our study, we utilized this methodology for both the initial cubic photonic crystal in SG #230 and the advanced tetragonal modulated gyrotropic photonic crystals in MSG #2.4. Further generalizations of this method are elaborated in our separate publication [36]. The process described in the aforementioned reference comprises three primary steps:

First, one starts by performing a symmetry analysis of the electromagnetic field profile in the UC at high-symmetry points, computing the symmetry eigenvalues ($x_{n,\mathbf{k}}$) for the relevant modes (\mathbf{k}) and bands (n) of interest and for each of the symmetry operation (g) in the SG:

$$x_{n,\mathbf{k}}(g) = \langle \mathbf{E}_{n,\mathbf{k}_h} | g \mathbf{D}_{n,\mathbf{k}_h} \rangle_{\text{UC}} \quad (8)$$

where the integral is performed within the UC. From the symmetry content thus obtained, one should extrapolate the photonic pseudo-orbitals needed for the construction of the photonic TB: This step can be done systematically for any SG, applying the algorithm described in Ref. [36]. The algorithm employs Topological Quantum Chemistry [43] to extract the orbital content of the bands of interest. As a third step, one employs these orbitals to construct a TB Hamiltonian ($H(\mathbf{k})$): the model's invariance under all SG symmetry transformations (g) is ensured by imposing the following constraint:

$$gH(\mathbf{k})g^{-1} = H(g\mathbf{k}). \quad (9)$$

After spectral filtering of the transversal modes from the longitudinal ones, also detailed in the same reference, one obtains the TETB for the photonic bands. Importantly, any step of this method can be applied to any arbitrary SG and UC of interest.

3) The following references could be added to reflect the state-of-the-art.

<https://www.nature.com/articles/s41467-022-30909-0>

<https://www.sciencedirect.com/science/article/abs/pii/S0010465522002120>

<https://journals.aps.org/prb/abstract/10.1103/PhysRevB.99.235423>

<https://journals.aps.org/prb/abstract/10.1103/PhysRevB.101.195105>

<https://onlinelibrary.wiley.com/doi/full/10.1002/adom.201900900>

<https://www.degruyter.com/document/doi/10.1515/nanoph-2019-0451/html>

We appreciate the referee sharing these references, which we found relevant for enhancing the state-of-the-art of our work, in relation to higher-order topological photonics [44]. As well, we thank the referee for highlighting Refs. [45] and [46], which present a coupled-dipole method for studying boundary responses of finite photonic systems. Since they offer insights into alternative approaches to tight-binding models in photonics, we have included these references to enrich the state-of-the-art of the methods section.

-
- * chiara.devescovi@dipc.org
† aitzolgarcia@dipc.org
‡ maia.vergniory@cpfs.mpg.de
- [1] Clarice D. Aiello, Muneer Abbas, John M. Abendroth, Andrei Afanasev, Shivang Agarwal, Amartya S. Banerjee, David N. Beratan, Jason N. Belling, Bertrand Berche, Antia Botana, et al. A chirality-based quantum leap: A forward-looking review. *ACS Nano*, 16(4):4989–5035, 2022.
 - [2] Ashwin C Atre, Aitzol García-Etxarri, Hadiseh Alaeian, and Jennifer A Dionne. A broadband negative index metamaterial at optical frequencies. *Advanced Optical Materials*, 1(4):327–333, 2013.
 - [3] Ashwin C Atre, Benjamin JM Brenny, Toon Coenen, Aitzol García-Etxarri, Albert Polman, and Jennifer A Dionne. Nanoscale optical tomography with cathodoluminescence spectroscopy. *Nature nanotechnology*, 10:429–436, 2015.
 - [4] Aitzol Garcia-Etxarri and Jennifer A Dionne. Surface enhanced circular dichroism spectroscopy mediated by non-chiral nanoantennas. *Physical Review B*, 87:235409, 2013.
 - [5] Aitzol García-Etxarri, Jesus M Ugalde, Juan Jose Sáenz, and Vladimiro Mujica. Field-mediated chirality information transfer in molecule–nanoparticle hybrids. *The Journal of Physical Chemistry C*, 124(2):1560–1565, 2019.
 - [6] Eric Sidney Aaron Goerlitzer, Mario Zapata-Herrera, Ekaterina Ponomareva, Deborah Feller, Aitzol Garcia-Etxarri, Matthias Karg, Javier Aizpurua, and Nicolas Vogel. Molecular-induced chirality transfer to plasmonic lattice modes. *ACS photonics*, 2023.
 - [7] Chi-Sing Ho, Aitzol Garcia-Etxarri, Yang Zhao, and Jennifer Dionne. Enhancing enantioselective absorption using dielectric nanospheres. *ACS Photonics*, 4(2):197–203, 2017.
 - [8] Jon Lasar-Alonso, Diego R Abujetas, Alvaro Nodar, Jennifer A Dionne, Juan Jose Saenz, Gabriel Molina-Terriza, Javier Aizpurua, and Aitzol Garcia-Etxarri. Surface-enhanced circular dichroism spectroscopy on periodic dual nanostructures. *ACS Photonics*, 7(11):2978–2986, 2020.
 - [9] Jon Lasar-Alonso, Jorge Olmos-Trigo, Chiara Devescovi, Pilar Hernández, Aitzol García-Etxarri, and Gabriel Molina-Terriza. Resonant helicity mixing of electromagnetic waves propagating through matter. *Physical Review Research*, 5(2):023116, 2023.
 - [10] Jorge Olmos-Trigo, Cristina Sanz-Fernández, Aitzol García-Etxarri, Gabriel Molina-Terriza, F Sebastián Bergeret, and Juan José Sáenz. Enhanced spin-orbit optical mirages from dual nanospheres. *Physical Review A*, 99(1):013852, 2019.
 - [11] Jorge Olmos-Trigo, Cristina Sanz-Fernández, Diego R Abujetas, Jon Lasar-Alonso, Nuno de Sousa, Aitzol García-Etxarri, José A Sánchez-Gil, Gabriel Molina-Terriza, and Juan José Sáenz. Kerker conditions upon lossless, absorption, and optical gain regimes. *Physical Review Letters*, 125:073205, 2020.
 - [12] Lisa V Poulikakos, Jennifer A Dionne, and Aitzol García-Etxarri. Optical helicity and optical chirality in free space and in the presence of matter. *Symmetry*, 11(9):1113, 2019.
 - [13] Sassan Sheikholeslami, Aitzol García-Etxarri, and Jennifer A Dionne. Controlling the interplay of electric and magnetic modes via fano-like plasmon resonances. *Nano letters*, 11(9):3927–3934, 2011.
 - [14] Michelle L Solomon, Jack Hu, Mark Lawrence, Aitzol García-Etxarri, and Jennifer A Dionne. Enantiospecific optical enhancement of chiral sensing and separation with dielectric metasurfaces. *ACS Photonics*, 6(1):43–49, 2018.
 - [15] A Martín-Ruiz, M Cambiaso, and LF Urrutia. The magnetoelectric coupling in electrodynamics. *International Journal of Modern Physics A*, 34(28):1941002, 2019.
 - [16] Yuri N Obukhov and Friedrich W Hehl. Measuring a piecewise constant axion field in classical electrodynamics. *Physics Letters A*, 341(5-6):357–365, 2005.
 - [17] Andreas Karch. Electric-magnetic duality and topological insulators. *Physical review letters*, 103(17):171601, 2009.
 - [18] Jonathan Gratus, Martin W McCall, and Paul Kinsler. Electromagnetism, axions, and topology: a first-order operator approach to constitutive responses provides greater freedom. *Physical Review A*, 101(4):043804, 2020.
 - [19] Yuanpei Lan, Shaolong Wan, and Shou-Cheng Zhang. Generalized quantization condition for topological insulators. *Physical Review B*, 83(20):205109, 2011.
 - [20] Alexander B Khanikaev, S Hossein Mousavi, Wang-Kong Tse, Mehdi Kargarian, Allan H MacDonald, and Gennady Shvets. Photonic topological insulators. *Nature materials*, 12(3):233–239, 2013.
 - [21] Xiaojun Cheng, Camille Jouvaud, Xiang Ni, S Hossein Mousavi, Azriel Z Genack, and Alexander B Khanikaev. Robust reconfigurable electromagnetic pathways within a photonic topological insulator. *Nature materials*, 15(5):542–548, 2016.
 - [22] Kueifu Lai, Tsuhsuang Ma, Xiao Bo, Steven Anlage, and Gennady Shvets. Experimental realization of a reflections-free compact delay line based on a photonic topological insulator. *Scientific reports*, 6(1):28453, 2016.
 - [23] Alexey Slobozhanyuk, S Hossein Mousavi, Xiang Ni, Daria Smirnova, Yuri S Kivshar, and Alexander B Khanikaev. Three-dimensional all-dielectric photonic topological insulator. *Nature Photonics*, 11(2):130–136, 2017.
 - [24] Daniel A Jacobs, Andrey E Miroshnichenko, Yuri S Kivshar, and Alexander B Khanikaev. Photonic topological chern

- insulators based on tellegen metacrystals. *New Journal of Physics*, 17(12):125015, 2015.
- [25] Filipa R Prudêncio and Mário G Silveirinha. Synthetic axion response with space-time crystals. *Physical Review Applied*, 19(2):024031, 2023.
- [26] João C Serra and Mário G Silveirinha. Engineering topological phases with a traveling-wave spacetime modulation. *arXiv preprint arXiv:2309.15320*, 2023.
- [27] Gui-Geng Liu, Zhen Gao, Qiang Wang, Xiang Xi, Yuan-Hang Hu, Maoren Wang, Chengqi Liu, Xiao Lin, Longjiang Deng, Shengyuan A Yang, et al. Topological chern vectors in three-dimensional photonic crystals. *Nature*, 609(7929):925–930, 2022.
- [28] Nicodemos Varnava and David Vanderbilt. Surfaces of axion insulators. *Physical Review B*, 98(24):245117, 2018.
- [29] Nicodemos Varnava, Ivo Souza, and David Vanderbilt. Axion coupling in the hybrid wannier representation. *Physical Review B*, 101(15):155130, 2020.
- [30] Luis Elcoro, Benjamin J Wieder, Zhida Song, Yuanfeng Xu, Barry Bradlyn, and B Andrei Bernevig. Magnetic topological quantum chemistry. *Nature communications*, 12(1):1–10, 2021.
- [31] Yuanfeng Xu, Luis Elcoro, Zhi-Da Song, Benjamin J Wieder, MG Vergniory, Nicolas Regnault, Yulin Chen, Claudia Felser, and B Andrei Bernevig. High-throughput calculations of magnetic topological materials. *Nature*, 586(7831):702–707, 2020.
- [32] Ling Lu, Haozhe Gao, and Zhong Wang. Topological one-way fiber of second chern number. *Nature communications*, 9(1):5384, 2018.
- [33] Eslam Khalaf. Higher-order topological insulators and superconductors protected by inversion symmetry. *Physical Review B*, 97(20):205136, 2018.
- [34] Anatoliui Konstantinovich Zvezdin and Viacheslav Alekseevich Kotov. *Modern magneto-optics and magneto-optical materials*. CRC Press, 1997.
- [35] Lev Davidovich Landau, John Stewart Bell, MJ Kearsley, LP Pitaevskii, EM Lifshitz, and JB Sykes. *Electrodynamics of continuous media*, volume 8. elsevier, 2013.
- [36] Antonio Morales-Pérez, Chiara Devescovi, Yoonseok Hwang, Mikel García-Díez, Barry Bradlyn, Juan Luis Mañes, Maia G Vergniory, and Aitzol García-Etxarri. Transversality-enforced tight-binding model for 3d photonic crystals aided by topological quantum chemistry. *arXiv preprint arXiv:2305.18257*, 2023.
- [37] Zhi-Da Song, Biao Lian, Raquel Queiroz, Roni Ilan, B Andrei Bernevig, and Ady Stern. Delocalization transition of a disordered axion insulator. *Physical review letters*, 127(1):016602, 2021.
- [38] Yang Linyun, Xi Xiang, Meng Yan, Zhu Zhenxiao, Wu Ying, Chen Jingming, Cheng Minqi, Xiang Kexin, Shum Perry Ping, Yang Yihao, et al. Acoustic three-dimensional chern insulators with arbitrary chern vectors. *arXiv preprint arXiv:2401.07040*, 2024.
- [39] Yujiang Ding, Yugui Peng, Yifan Zhu, Xudong Fan, Jing Yang, Bin Liang, Xuefeng Zhu, Xiangang Wan, and Jianchun Cheng. Experimental demonstration of acoustic chern insulators. *Physical Review Letters*, 122(1):014302, 2019.
- [40] Zhaoju Yang, Fei Gao, Xihang Shi, Xiao Lin, Zhen Gao, Yidong Chong, and Baile Zhang. Topological acoustics. *Physical review letters*, 114(11):114301, 2015.
- [41] Xu Ni, Cheng He, Xiao-Chen Sun, Xiao-ping Liu, Ming-Hui Lu, Liang Feng, and Yan-Feng Chen. Topologically protected one-way edge mode in networks of acoustic resonators with circulating air flow. *New Journal of Physics*, 17(5):053016, 2015.
- [42] Alexander B Khanikaev, Romain Fleury, S Hossein Mousavi, and Andrea Alu. Topologically robust sound propagation in an angular-momentum-biased graphene-like resonator lattice. *Nature communications*, 6(1):8260, 2015.
- [43] Barry Bradlyn, Luis Elcoro, Jennifer Cano, Maia G Vergniory, Zhijun Wang, Claudia Felser, Mois I Aroyo, and B Andrei Bernevig. Topological quantum chemistry. *Nature*, 547(7663):298–305, 2017.
- [44] Minkyung Kim and Junsuk Rho. Topological edge and corner states in a two-dimensional photonic su-schrieffer-heeger lattice. *Nanophotonics*, 9(10):3227–3234, 2020.
- [45] Minkyung Kim and Junsuk Rho. Quantum hall phase and chiral edge states simulated by a coupled dipole method. *Physical Review B*, 101(19):195105, 2020.
- [46] Minkyung Kim and Junsuk Rho. Cdpds: Coupled dipole method-based photonic dispersion solver. *Computer Physics Communications*, 282:108493, 2023.

REVIEWERS' COMMENTS

Reviewer #1 (Remarks to the Author):

This reviewer appreciates tremendous efforts taken by the authors to revise the manuscript. I would say that this is a truly exemplary case of properly addressed reviewers' criticism. This manuscript now is much more focused on physics and the significance of axion topology which makes it accessible to a broader readership. I recommend publication of this work in its present form.

Reviewer #2 (Remarks to the Author):

I have read the replies to the referees and the revised article. The authors addressed my questions satisfactorily. I recommend this paper for publication in Nature Communications in its current form.

Reviewer #3 (Remarks to the Author):

The authors revised the manuscript reflecting all the queries and comments. Comparing the initial submission, the quality, clarity and presentation of the work improved a lot. I recommend this manuscript to be published in Nature Communications.

Axion Topology in Photonic Crystal Domain Walls: Response to the reviewers' comments

Chiara Devescovi^{*,1,*} Antonio Morales-Pérez^{*,1} Yoonseok Hwang,² Mikel García-Díez,^{1,3} Iñigo Robredo,^{4,1} Juan Luis Mañes,³ Barry Bradlyn,² Aitzol García-Etxarri,^{1,†} and Maia G. Vergniory^{4,1,‡}

¹*Donostia International Physics Center, Paseo Manuel de Lardizabal 4, 20018 Donostia-San Sebastian, Spain.*

²*Department of Physics, University of Illinois at Urbana-Champaign, Urbana, IL, USA*

³*Physics Department, University of the Basque Country (UPV/EHU), Bilbao, Spain*

⁴*Max Planck Institute for Chemical Physics of Solids, Dresden D-01187, Germany*

(Dated: June 13, 2024)

Kind referees,

We thank you again for your valuable feedback and for acknowledging the efforts put into revising the manuscript. We are grateful for the constructive comments, during the revision process, which have significantly improved the clarity and reach of our work. We appreciate your final recommendations for publication.

Best wishes,

Chiara Devescovi and Aitzol Garcia-Etxarri on behalf of all authors

Reviewer No.1:

This reviewer appreciates tremendous efforts taken by the authors to revise the manuscript. I would say that this is a truly exemplary case of properly addressed reviewers' criticism. This manuscript now is much more focused on physics and the significance of axion topology which makes it accessible to a broader readership. I recommend publication of this work in its present form.

Reviewer No. 2:

I have read the replies to the referees and the revised article. The authors addressed my questions satisfactorily. I recommend this paper for publication in Nature Communications in its current form.

Reviewer No. 3:

The authors revised the manuscript reflecting all the queries and comments. Comparing the initial submission, the quality, clarity and presentation of the work improved a lot. I recommend this manuscript to be published in Nature Communications.

* chiara.devescovi@dipc.org

† aitzolgarcia@dipc.org

‡ maia.vergniory@cpfs.mpg.de